# Monolithic FAPbBr$_3$ photoanode for photoelectrochemical water oxidation with low onset-potential and enhanced stability

Hao Yang[1,5], Yawen Liu[2,5], Yunxuan Ding[3], Fusheng Li [4], Linqin Wang[3], Bin Cai[2], Fuguo Zhang[1], Tianqi Liu [1], Gerrit Boschloo[2], Erik M. J. Johansson [2] & Licheng Sun [1,3,4]

Despite considerable research efforts on photoelectrochemical water splitting over the past decades, practical application faces challenges by the absence of efficient, stable, and scalable photoelectrodes. Herein, we report a metal-halide perovskite-based photoanode for photoelectrochemical water oxidation. With a planar structure using mesoporous carbon as a hole-conducting layer, the precious metal-free FAPbBr$_3$ photovoltaic device achieves 9.2% solar-to-electrical power conversion efficiency and 1.4 V open-circuit voltage. The photovoltaic architecture successfully applies to build a monolithic photoanode with the FAPbBr$_3$ absorber, carbon/graphite conductive protection layers, and NiFe catalyst layers for water oxidation. The photoanode delivers ultralow onset potential below 0 V versus the reversible hydrogen electrode and high applied bias photon-to-current efficiency of 8.5%. Stable operation exceeding 100 h under solar illumination by applying ultraviolet-filter protection. The photothermal investigation verifies the performance boost in perovskite photoanode by photothermal effect. This study is significant in guiding the development of photovoltaic material-based photoelectrodes for solar fuel applications.

Artificial photosynthesis is a system that uses artificial materials to mimic the conversion of solar energy to chemical energy in nature. As solar energy is the most abundant energy source on Earth, using water as a raw material to produce hydrogen with solar energy is an ideal approach to obtaining renewable fuels, known as the solar fuel pathway[1–3]. Developing efficient, stable, and low-cost water-splitting devices is one of the key scientific challenges to achieving large-scale electrolysis and photolysis of water for hydrogen production. Currently, there are two main artificial photosynthesis approaches for converting solar energy into fuels: one is the indirect approach, which uses the electricity generated by a photovoltaic (PV) device to drive an electrochemical (EC) reaction to produce hydrogen; the other is the direct approach which takes inspiration from the working principles of natural photosynthesis, and converts solar energy into hydrogen and oxygen through the photoelectrochemical (PEC) water splitting without external power[4].

Figure 1c lists the band structure of typical materials for PEC water splitting. Although visible (vis.) light-absorbing materials, including WO$_3$, BiVO$_4$, and Fe$_2$O$_3$, have been extensively studied due to favorable band edge positions, their limited charge separation/transport properties remain major obstacles to achieving commercially viable performance[5]. Most PEC cells with single-junction semiconductor

[1]Department of Chemistry, School of Engineering Sciences in Chemistry, Biotechnology and Health, KTH Royal Institute of Technology, 10044 Stockholm, Sweden. [2]Department of Chemistry-Ångström, Physical Chemistry, Uppsala University, 75120 Uppsala, Sweden. [3]Center of Artificial Photosynthesis for Solar Fuels and Department of Chemistry, School of Science, Westlake University, 310024 Hangzhou, China. [4]State Key Laboratory of Fine Chemicals, Institute of Artificial Photosynthesis, DUT-KTH Joint Education and Research Centre on Molecular Devices, Dalian University of Technology, 116024 Dalian, China. [5]These authors contributed equally: Hao Yang, Yawen Liu. ✉e-mail: erik.johansson@kemi.uu.se; lichengs@kth.se

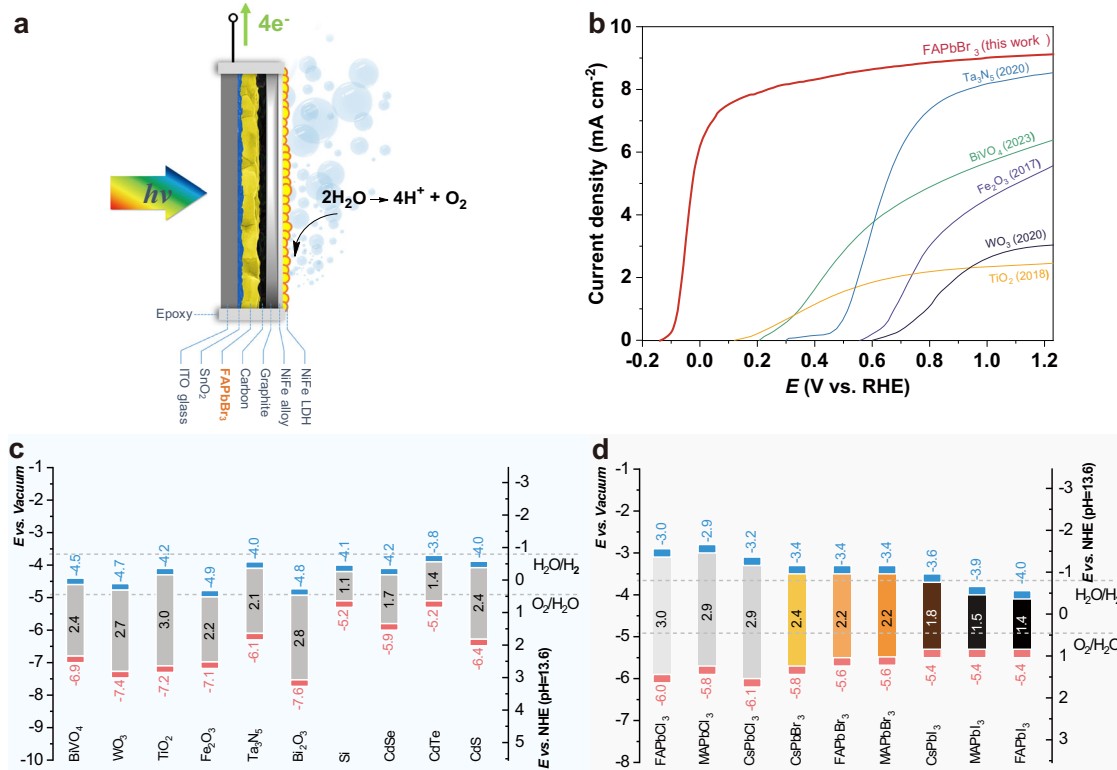

**Fig. 1 | Comparing band structures and photoanode performance. a** Schematic illustration of FAPbBr$_3$ photoanode for oxygen evolution. **b** Comparison of the linear sweep voltammetry (LSV) curves between the FAPbBr$_3$ photoanode and the best-reported photoanodes made from a single junction absorber layer, including TiO$_2$, WO$_3$, Fe$_2$O$_3$, BiVO$_4$, and Ta$_3$N$_5$, with catalysts layer. The LSV data was extracted from the corresponding literature. Band positions of (**c**) typical PEC and (**d**) lead halide perovskite semiconductors in the pH 13.6 aqueous electrolyte compared with the energy potential for water splitting reaction. red=valence band edge; blue=conduction band edge. Tabulated values are retrieved from references[7,18,61].

photoelectrodes, even those made from wide band gap ($E_g$) semiconductor materials such as TiO$_2$ and WO$_3$, show high current onset potential due to the high thermodynamic requirements and the multi-proton and electron transfer processes involved in water oxidation reactions. As such, the development of a perfect semiconductor that is stable, low-cost, has suitable band structures, high charge carrier mobility, and high absorption coefficient remains a considerable challenge. At the same time, better water oxidation catalysts, as well as reliable, stable, and economical catalyst loading techniques, are also highly desired to prepare high-efficiency water oxidation photoanodes.

Alternatively, the PV material (Si, GaAs, GaInP) with excellent optical and electronic properties can be arranged as an absorber layer in an integrated solar hydrogen device, analog as the semiconductor layer in traditional PEC configuration. Among PV absorbers, lead halide perovskite (ABX$_3$) materials have emerged as a promising alternative to overcome the limitations of metal-oxide-based photoelectrodes due to their exceptional optoelectronic properties[6]. These materials offer several advantages over traditional metal-oxide-based photoelectrodes, including higher light absorption coefficients, facile exciton dissociation ability, and longer carrier diffusion lengths[7]. Additionally, their unique crystal structure enables facile tuning of their electronic properties through compositional variations, making them an attractive platform for the development of high-performance photoelectrochemical systems. As such, perovskite materials are a subject of intense research and development in the field of solar fuel production[8–10]. Lead halide perovskite materials with iodine components have a high theoretical saturation current due to their narrow band gap (Fig. 1d). However, it is challenging for a single cell to generate enough photoelectric voltage for water oxidation. Therefore,

when used as a single junction photoanode, their onset potential is typically greater than 0.5 V vs. reversible hydrogen electrode (RHE) (Supplementary Fig. 2). Meanwhile, the performance of chloride perovskites is limited by their wide band gap, which leads to extremely low visible light absorption. Therefore, bromide perovskites with suitable energy band structures are ideal components for the preparation of low-onset potential photoanodes. Out of the three mono-cation bromide perovskites, cesium lead bromide (CsPbBr$_3$) has the larger band gap at 2.4 eV, i.e., the lowest theoretical saturation current. Additionally, its inorganic counterparts require high-temperature annealing over 300 °C for stabilization, which makes it difficult to obtain high-crystallinity films[11]. On the other hand, methylammonium lead bromide (MAPbBr$_3$) components have poor thermal stability and are not suitable for long-term high-temperature operations. Therefore, after a comprehensive consideration of the preparation difficulty, band structure, and material stability, formamidinium lead bromide (FAPbBr$_3$) is selected in this study to be the top choice for producing low-onset potential photoanodes. The photovoltaic materials-based photoelectrochemical (PVM-PEC) cell can immerse in an electrolyte after coating with waterproof layers[12]. Among the strategies reported, Poli et al. utilize a mesoporous carbon/graphite sheet structure as protective layers on a CsPbBr$_3$-based photoanode[13]. This simple, cost-effective encapsulation method provides stable protection of the absorptive layer against moisture, while maintaining high conductivity. This approach offers promising opportunities for the use of water-sensitive perovskite materials in integrated PEC cells.

Herein, we present the first efficient and stable FAPbBr$_3$-based photoanode for solar water oxidation. This FAPbBr$_3$ absorber was initially employed in mesoporous carbon solar cells, yielding efficiencies of 9.16% and open-circuit voltages of up to 1.4 V. Furthermore,

the $FAPbBr_3$ absorber, after encapsulated with graphite and resin/epoxy, was utilized to produce a photoanode for water oxidation (Fig. 1a). The optimized photoanode with integrated carbon/graphite/alloy interlayers and electrodeposited NiFe layered double hydroxide (LDH) catalyst subsequently demonstrates a remarkable performance and stability: the device outperforms oxide photoanodes by exhibiting an ultralow onset potential below 0 V vs. RHE and achieving a maximum applied bias photon-to-current efficiency (ABPE) of 8.5%. The performance of $FAPbBr_3$ system far exceeds that of the best-reported single-junction absorber photoanodes to date (Fig. 1b), including $TiO_2$[14], $WO_3$[15], $Fe_2O_3$[16], $BiVO_4$[17], and $Ta_3N_5$[18]. Additionally, it maintains 95% of its performance for 100 h of operation, surpassing that of all known reported perovskite photoanodes. We further discuss the ultraviolet (UV) degradation of the perovskite interface and emphasize the feasibility of utilizing the photothermal effect to enhance cell performance. The as-fabricated $FAPbBr_3$ photoanode is produced without the need for high temperature, inert atmosphere, or high vacuum evaporation techniques, providing promising opportunities for the development of large scale high-performance, low-cost perovskite photoanodes for water oxidation.

## Results

### Photoanode preparation and characterization

As illustrated in Fig. 2a, the wide band gap $FAPbBr_3$ perovskite photoanode consists of two parts, the light-absorbing part ($ITO/SnO_2/FAPbBr_3/carbon$) and the water oxidation catalyst (WOC) part (graphite sheet/NiFe alloy/NiFe LDH); the edge of the device is sealed with epoxy resin to prevent water degradation of $FAPbBr_3$ material (Supplementary Fig. 1). The light-absorbing part, which is a hole transport material (HTM)-free $FAPbBr_3$ solar cell, exhibits outstanding performance with a power conversion efficiency (PCE) of 9.16%, a high open-circuit voltage ($V_{oc}$) of 1.38 V, a short-circuit current density ($J_{sc}$) of 8.69 mA cm$^{-2}$, and a fill factor (FF) of 76.04% (see Supplementary Note 1 for details). Inspired by the work of Poli et al. [13], a commercially available 160 μm self-adhesive graphite sheet (GS) was placed on top of the $FAPbBr_3$/carbon layer to prevent water penetration into the light-absorbing part. As demonstrated in Supplementary Fig. 17, for the epoxy resin-encapsulated device with solely the mesoporous carbon layer, notable degradation was observed after just 5 min of water immersion. Conversely, the device protected by the GS layer exhibited minimal alterations even after 100 h of water immersion. The compact and stacked GS separator offers excellent thermal (350 W m$^{-1}$ K$^{-1}$) and electrical conductivity while also providing more efficient protection for the perovskite from liquid water percolation. Moreover, the self-adhesive and commercially available characteristics of the GS simplify the device fabrication process, making it an ideal water separator component for the assembly of a perovskite photoanode. By utilizing facile electrodeposition methods, the WOCs could be attached to the conductive GS layer, which also allows individual optimization of the WOC component on the indium tin oxide (ITO)/GS substrate. The overpotential requirements of the as-fabricated GS/NiFe alloy/NiFe LDH electrode to reach a constant current density of 10 mA cm$^{-2}$ is only 228 mV; the overpotential shows only an increment of 19 mV after 100 h of catalysis (see Supplementary Note 2 for details).

### Photoelectrochemical performance of photoanode

After conducting detailed studies on the performance, stability, morphology, and structure of the WOC part, highly active GS/NiFe alloy/NiFe LDH layers were applied to the light-absorbing part to construct a photoanode with an architecture of $ITO/SnO_2/FAPbBr_3/carbon/GS/NiFe$ alloy/NiFe LDH (i.e., $FAPbBr_3$ photoanode). The digital images of $FAPbBr_3$ photoanode are illustrated in Fig. 2a. Fig. 2b shows the LSV curve of the as-fabricated $FAPbBr_3$ photoanode; the photocurrent sharply rose from −0.12 V vs. RHE and reached a saturated photocurrent density over 8.5 mA cm$^{-2}$ at 1.23 V vs. RHE. The inset figures

provide statistical data of 20 devices, indicating that the average onset potential is −0.066 V vs. RHE, and the average photocurrent density at 1.23 V vs. RHE is approximately 8.7 mA cm$^{-2}$. The photoanode exhibits a hysteresis effect similar to a solar cell, with a decay of about 50 mV on the back scanning of the onset potential; additionally, the cyclic voltammetry (CV) curves show almost no scan rate dependency (Supplementary Fig. 18). As shown in Fig. 2d, the incident photon-to-current efficiencies (IPCEs) of $FAPbBr_3$ photoanode were measured and calculated under different potentials. The IPCE values are approximately 80 % and 90 % at 0.23 and 1.23 V vs. RHE, respectively. Integrating the IPCE curves over the AM 1.5 G solar spectrum, the theoretical photocurrent densities are estimated to be 7.4 mA cm$^{-2}$ at 0.23 V vs. RHE and 8.3 mA cm$^{-2}$ at 1.23 V vs. RHE. These calculated values are consistent with the measured data in Fig. 2b, which are 7.8 mA cm$^{-2}$ at 0.23 V and 8.5 mA cm$^{-2}$ at 1.23 V. The $FAPbBr_3$ photoanode exhibits an ultralow overpotential and over 8 mA cm$^{-2}$ saturation current, making it a perfect match with photocathodes, such as CuO or $GaInP_2$, which have a saturation current of around 10 mA cm$^{-2}$[19,20]. When subjected to chopped-light illumination, the photoanode exhibits notable current spikes around onset potentials in both LSV and chronoamperometry current due to the slow water oxidation kinetics induced carrier recombination on the surface (Fig. 2c and Supplementary Fig. 19). The excess charge may be dissipated through radiative recombination, which can be detrimental to the device's lifetime. The spikes are surpassed when the potential increases to 0.2 V vs. RHE, indicating that only a subtle additional driving force is needed for the $FAPbBr_3$ photoanode to improve charge separation and catalytic efficiency, which greatly decrease the onset potential requirements of the counter photocathodes in Z-scheme PEC cells. The steady-state current density of the $FAPbBr_3$ photoanode is also consistent with LSV current density; under the applied potentials of 0.2 and 1.2 V vs. RHE, steady current densities of 7.20 and 8.46 mA cm$^{-2}$ were achieved, respectively (Supplementary Fig. 19). The amount of photogenerated $O_2$ in the headspace was quantified using a pressure transducer. The number of electrons passing through the electrode agreed well with the amount of $O_2$ detected, representing Faradaic efficiencies of 94.7% and 96.2 % at the potential of 0.23 and 1.23 V vs. RHE, respectively (Supplementary Fig. 20).

The ABPE was calculated from the corresponding LSV curve in a three-electrode system (Supplementary Fig. 21). A maximum ABPE of 8.52 % is achieved at 0.082 V vs. RHE. This ABPE value is compared to those reported systems in Fig. 2e. To the best of our knowledge, the value is one of the best of those reported photoanodes, including metal oxide, perovskite, polymer bulk heterojunction (BHJ), and silicon-based photoanodes. Compared to bismuth vanadate-based photoanodes, which have a similar wide $E_g$, the $FAPbBr_3$ photoanode is far superior in terms of maximum ABPE. Moreover, considering the ultralow potential at which the maximum ABPE value is achieved, the $FAPbBr_3$ photoanode is one of the most promising candidates for realizing efficient photo-driven total water splitting. Fig. 2f summarizes the best-reported photovoltages for representative PEC and PV devices with varying band gaps. The photovoltage of PV devices is presented as the open circuit potential, whereas the photovoltage of photoanodes is calculated as the difference between the photocurrent onset potential and the Nernstian potential for water oxidation (1.23 V vs. RHE)[21]. Analog to trends of PV devices, a wide band gap absorber (>2.0 eV) should be capable of photovoltages approaching >1.4 V when considering a free energy loss of 0.6 eV[21], while it is clear that traditional oxide or nitride wide band gap absorbers have significant room for photovoltage improvement to reach the goal. To the best of our knowledge, the wide band gap, HTM-free $FAPbBr_3$ photoanode presented in this work has achieved the lowest onset potential (around −0.12 V) and record photovoltage (around 1.35 V) among single junction photoanodes to date, which brings the photovoltage much closer to the optimal value (>1.6 V). The ultralow onset potential greatly

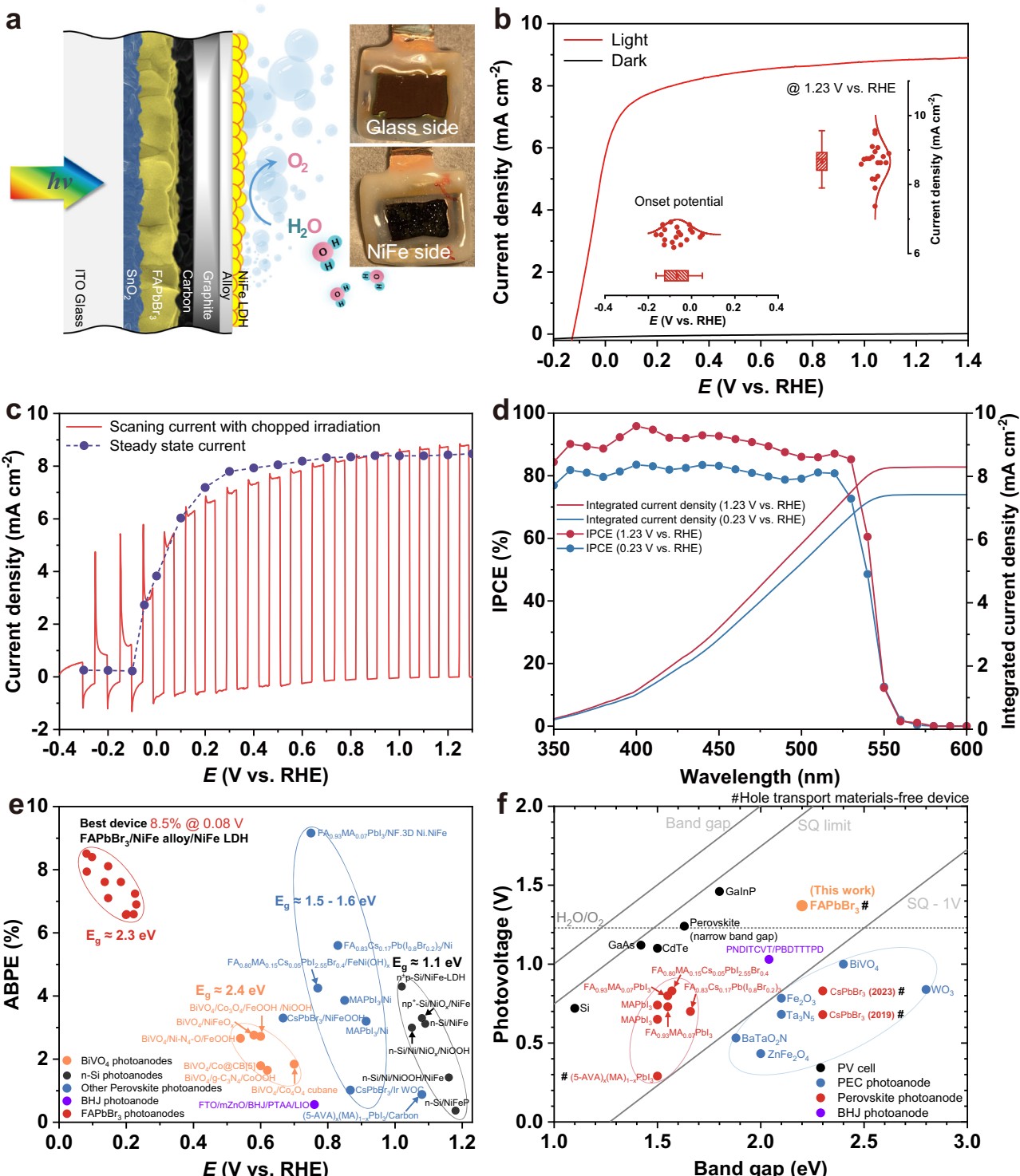

**Fig. 2 | Assessing and comparing FAPbBr₃ photoanode performance. a** Device structure of FAPbBr₃ photoanode in this work. Inset: digital images of as-prepared photoanode. **b** LSV curve of FAPbBr₃ photoanode in 1.0 M KOH (scan rate: 50 mV s⁻¹). Inset: statistical photoelectrochemical parameters (20 devices) of the onset potential and saturated catalytic current density at 1.23 V vs. RHE. **c** LSV curve of FAPbBr₃ photoanode under chopped irradiation (red curve) and current density-voltage response of FAPbBr₃ photoanode obtained under steady-state current conditions (blue dots) in 1.0 M KOH. **d** IPCEs and corresponding integral current densities (AM 1.5 G) of FAPbBr₃ photoanode at 1.23 V and 0.23 V vs. RHE. **e** ABPE

benchmarks of as-prepared FAPbBr₃ photoanodes and other perovskite, BiVO₄, BHJ and silicon-based photoanodes. Tabulated values and literature references for listed photoanodes are provided in Supplementary Table 4. **f** Photovoltage benchmarks for PEC and PV materials as a function of the optical band gap. Diagonal lines represent the band gap, the Shockley-Queisser (S-Q) photovoltage limit, and the S-Q limit minus 1 V. Tabulated values of PV cells and PEC anodes were retrieved from reference[21]; Tabulated values and literature references for other perovskite and BHJ anodes are provided in the Supplementary Tables 5 and 6.

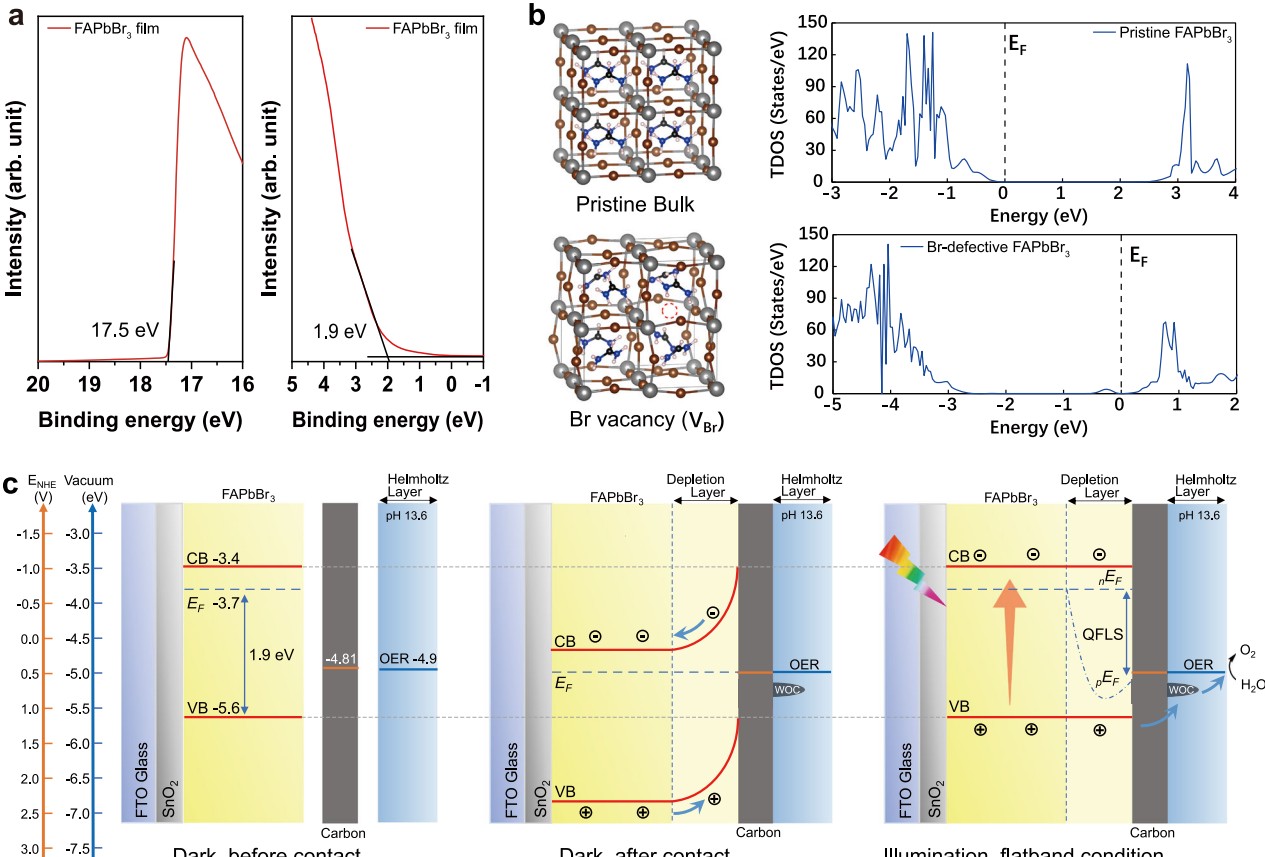

**Fig. 3 | Band structure analysis of FAPbBr₃ photoanode. a** UPS spectrum of FAPbBr₃ film. **b** Geometric structures and calculated TDOS of pristine and Br-defective FAPbBr₃ perovskites. **c** The band alignment structures of FAPbBr₃ photoanode under specific conditions.

reduces the requirements of the counter photocathode, making it more feasible to construct a low-cost Z-scheme water-splitting system with over 5% solar-to-hydrogen efficiency. Supplementary Fig. 22 shows the values of solar cell $V_{oc}$, photovoltage, and corresponding voltage loss of various perovskite photoanodes for water oxidation. Compared to all other reported photoanodes based on solar cell architecture, which exhibit a voltage loss of over 0.2 V, the FAPbBr₃ photoanode shows a significantly lower voltage loss of only 0.05 V. This low voltage loss suggests good internal contact and well-matched band structure in the device.

To gain deep insight into the origin of the ultralow onset potential, ultraviolet photoelectron spectroscopy (UPS) was used to explore the band alignments in devices (Fig. 3a). The Fermi level of the FAPbBr₃ film is measured to be approximately −3.7 eV vs. vacuum, and the valence band energy of the film was found to be 1.9 eV lower than the Fermi level (i.e., −5.6 eV vs. vacuum), indicating an n-type semiconductor feature of the FAPbBr₃ absorber layer. Density functional theory (DFT) calculations were performed to gain a detailed understanding of the energy states and n-type features of the as-fabricated FAPbBr₃ materials. The lattice parameters and band gap of pristine FAPbBr₃ were calculated to be 6.02 Å and 2.41 eV, respectively, which agreed well with the reported experiment values (6.01 Å and 2.3 eV)[22–24]. As perovskite is made through a two-step process, the vacancy state was considered to dominate the band structure due to non-stoichiometric components. Calculations were performed for both the pristine structure and possible vacancy structures, including A-site vacancy ($V_{FA}$), B-site vacancy ($V_{Pb}$), X-site vacancy ($V_{Br}$), B anti-site X ($Pb_{Br}$), and X anti-site B ($Br_{Pb}$). Fig. 3b presents the calculated total density of states (TDOS) of pristine and X-site vacancy perovskite

to reveal the energy band structure. Specifically, the projected density of states (PDOS) in Supplementary Fig. 23 revealed that the conduction band minimum (CBM) mainly consists of Pb 6p orbitals, while Br 4p orbitals contribute to the valence band maximum (VBM); moreover, the trap state energy level of $V_{Br}$ is close to the CBM, as it is composed of the Pb 6p orbitals. The Fermi energy level is found to be shifted to the CBM, indicating the n-type property of Br vacancy due to the presence of donor states within the band gap. The TDOS of other vacancy and anti-site structures were also calculated accordingly. As shown in Supplementary Fig. 24, $V_{Pb}$, $V_{FA}$, and $Br_{Pb}$ will reduce the Fermi level into the VBM, while $Pb_{Br}$ will shift the Fermi level penetrating into CBM. The defect formation energies (DFEs) for the aforementioned defect types were calculated according to the procedures in Supplementary Note 3. As detailed in Supplementary Table 7, the DFE for $Pb_{Br}$ was found to be 0.78 eV, which is higher than the −0.51 eV for the Br vacancy, which implies that $Pb_{Br}$ is less stable in comparison to $V_{Br}$. These theoretical results provide valuable insight into the effect of vacancy and anti-site structures on the Fermi level of FAPbBr₃ materials. Based on the results, it is suggested that the vacancy structure of Br leads to an n-type property of the FAPbBr₃ film.

The energetic situations at the n-type FAPbBr₃/electrolyte interface are presented in Fig. 3c. After the contact of the semiconductor surface with the electrolyte, the thermodynamic equilibrium on both sides of the interface is established with the formation of the depletion layer. After light illumination, the quasi-Fermi level for electrons is consistent with the Fermi level at equilibrium; in contrast, the quasi-Fermi level of holes shifts downwards to the vicinity of the valence band edge. In a PV device, the corresponding quasi-Fermi level splitting (QFLS) is directly related to the open circuit potential[25]. In this

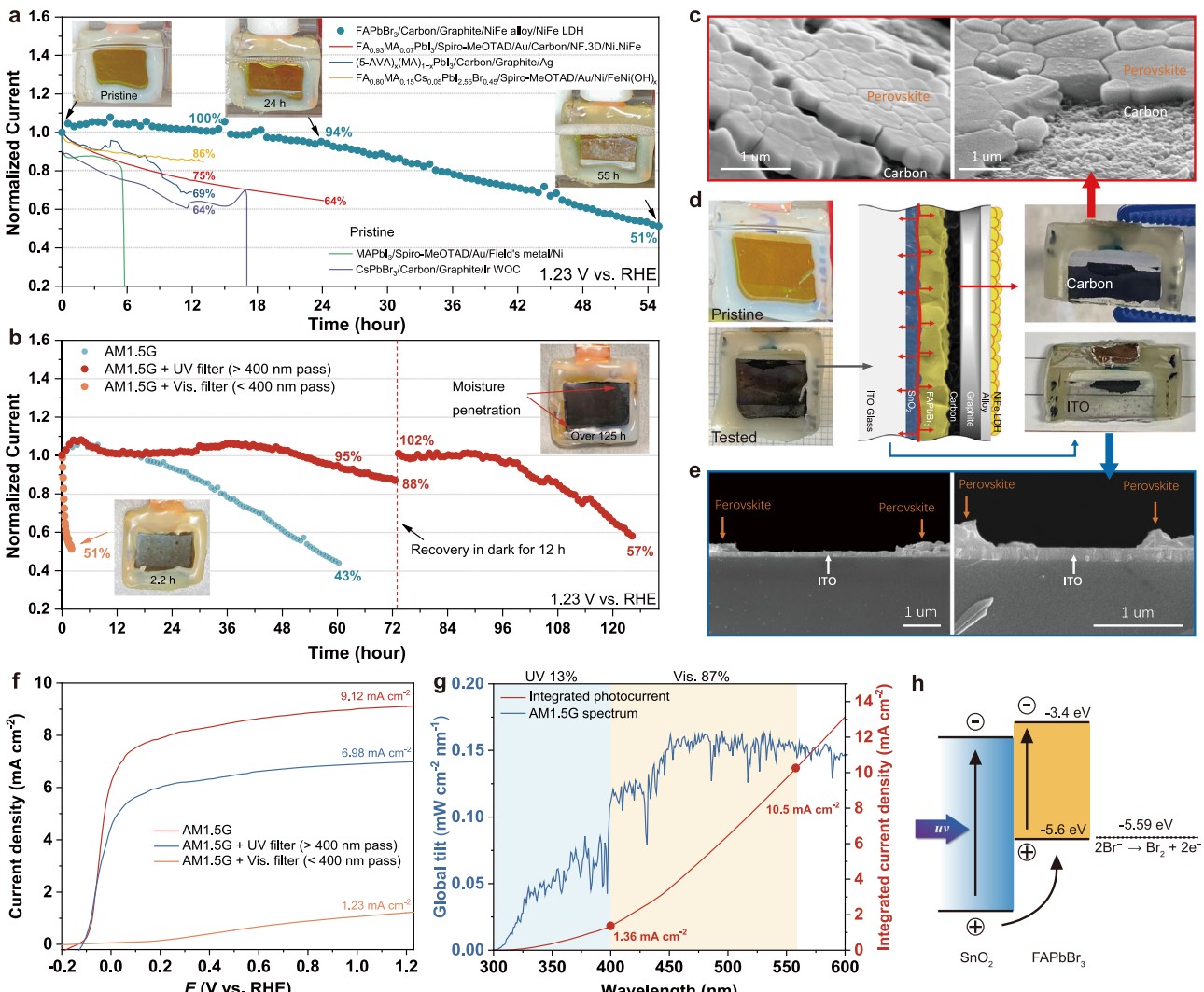

**Fig. 4 | Stability assessment of FAPbBr₃ photoanode. a** Normalized chronoamperometric measurement of FAPbBr₃ photoanode at 1.23 V vs. RHE in 1.0 M KOH solutions (AM 1.5 G, 100 mW cm⁻²). Inset: digital images of photoanode at different light-exposed times. **b** Normalized chronoamperometric measurement of FAPbBr₃ photoanode at 1.23 V vs. RHE with different filters in 1.0 M KOH solutions (AM 1.5 G + UV or Vis. filter). Inset: digital images of photoanode at different light-exposed conditions. **c** SEM images for the carbon part of tested FAPbBr₃ photoanode in oblique view. **d** Schematic diagram of the photoanode degradation process and digital images of pristine and tested FAPbBr₃ photoanode. **e** SEM images for the glass part of tested FAPbBr₃ photoanode in cross-section view. **f** LSV curves of FAPbBr₃ photoanode with different filters in 1.0 M KOH (scan rate: 50 mV s⁻¹). **g** AM 1.5 G spectrum and corresponding integral current densities for UV (< 400 nm) and maximum absorption edge (<560 nm) regions. **h** Schematic diagram of the proposed mechanism for UV degradation of FAPbBr₃ photoanode.

context, the QFLS of the FAPbBr₃ perovskite layer should not be less than 1.4 eV, as the perovskite layer can generate an open circuit potential of about 1.4 V in the carbon-based PV device. In the FAPbBr₃/ electrolyte system, the flat-band potential is estimated to be the Fermi level of the FAPbBr₃ film, which is −3.7 V vs. vacuum. The onset potential of photocurrent can be considered to coincide with the flat-band potential when the band bending is less pronounced[26]. In our case, the theoretical onset potential is $E_{onset} = (3.7 − 4.5) + 0.059 \times 13.6 = 0.02$ V vs. RHE. Meanwhile, with an QFLS over 1.4 eV, the oxidation potential of holes reaches over 1.42 V, which could offer a sufficient overpotential of 0.19 V for apparent water oxidation reaction. The theoretical value of $E_{onset}$ coincides with the measured $E_{onset}$ of −0.2 to 0.1 V vs. RHE in Fig. 2b, which confirms that the ultralow onset potential in the FAPbBr₃ photoanode is derived from the shallower Fermi level and higher photovoltage (see Supplementary Note 4 for detailed discussions). The above analysis highlights the crucial role of the Fermi level and band structure alignment of semiconductor materials in constructing high-performance, low-overpotential photoanode devices.

## Stability of photoanode

Although the NiFe alloy/NiFe LDH-based catalysts as well as HTM-free FAPbBr₃ solar cells have already shown good stability under operation conditions, achieving long-term stability using integrated photoelectrodes remains a challenging task due to the harsh electrolyte environment of water splitting. To investigate the long-term stability of FAPbBr₃ photoanodes, chronoamperometry with a constant applied potential was conducted in 1.0 M KOH solutions. As shown in Fig. 4a and Supplementary Fig. 27, the initial photocurrent exceeds 8.8 mA cm⁻² under the applied potential of 1.23 vs. RHE. The device exhibits a current half-life of over 55 h, which is better than other reported systems listed in Fig. 4a. Notably, the current shows no decay in the initial 12 h of illumination and remains at around 95% of its initial current after 24 h of operation. During the measurement, light-soaking-induced photocurrent enhancement is observed, where the photocurrent gradually increased with the initial 5 h of illumination[13,27]. The stability of the FAPbBr₃ photoanode was also examined under a lower applied potential of 0.3 vs. RHE. The current shows almost no decay after 18 h of illumination with an initial current density of

7.7 mA cm$^{-2}$ (Supplementary Fig. 28). The inset images in Fig. 4a show photographs of the PEC cell after 24 and 55 h of photo-electrolysis. With time increasing, the orange active area of the photoanode gradually turns white with the decayed photocurrent, indicating the gradual degradation of the perovskite layer inside the sealed PEC cell (not due to seal failure) since the degradation was only observed at the light illumination area (Supplementary Fig. S29a). The following investigation aimed to reveal possibilities for improving the stability of degraded photoanodes. To this end, deactivated photoanodes were disassembled to study the degradation mechanism. As shown in Fig. 4d, the tested sample could be easily separated into two parts, with the graphite/carbon layer peeled off from the glass substrate (see Supplementary Fig. 30 for details). Scanning electron microscope (SEM) was employed to observe the interface of the two peeled parts. As shown in Fig. 4c, the perovskite layer tightly adheres to the carbon layer; in contrast, only a small portion of perovskite crystal is attached to the ITO glass substrate (Fig. 4e). For the pristine sample, the ITO glass substrate should be an orange color with the growth of FAPbBr$_3$ layer. However, after stability tests, the FAPbBr$_3$ layer detached from the substrate, resulting in a transparent glass substrate. The detachment of the perovskite layer from the ITO substrate under working conditions was further confirmed in SEM analysis of selectively exposed photoanode (Supplementary Fig. 29c). The functional groups of the pristine and degraded FAPbBr$_3$ films were investigated using Fourier transform infrared (FT-IR) spectroscopy. As presented in Supplementary Fig. 31, a sharp peak at around 1716 cm$^{-1}$ is observed in the pristine FAPbBr$_3$ film, which is attributed to the antisymmetric C-N vibration; as a reference, the antisymmetric C-N peak of pure FABr is broad and centered at 1697 cm$^{-1}$. The antisymmetric C-N peak of degraded FAPbBr$_3$ film is found to be blue-shifted and broader than that of the pristine sample, indicating the evolution of ITO-SnO$_2$/FAPbBr$_3$ interface under light irradiation, with the possibility that the perovskite degrades into a corresponding monomer (i.e., FABr).

The stability of lead halide perovskite solar cells is influenced by several factors, including heat, moisture, voltage, and UV light. Among these factors, the stability upon UV light exposure is particularly problematic due to the photocatalytic effect of electron transport materials like TiO$_2$ and SnO$_2$, which are considered to be the main reasons for perovskite degradation[28]. The theoretical oxidation potential of bromine anion (Br$^-$) is 1.087 V vs. normal hydrogen electrode, which is equivalent to $-5.59$ eV vs. vacuum level[29]. The potential is almost the same as the VBM of FAPbBr$_3$ perovskite. Thus, under visible light irradiation, oxidation of Br$^-$ by excited holes is less likely to occur on FAPbBr$_3$ because such the reaction requires a certain additional driving force. However, as indicated in Fig. 4h, highly oxidative holes would accumulate close to the SnO$_2$/perovskite interface when UV light exposure. These charges could extract electrons from the Br$^-$ in the SnO$_2$/FAPbBr$_3$ junction, compromising the perovskite interface structure and producing FABr or PbBr$_2$[30]. To further confirm the negative impact of UV light, stability tests were conducted using a long-pass UV filter with a cut-off wavelength of >400 nm and a band-pass visible light filter with a range of 200–400 nm (Supplementary Fig. 32). As shown in Fig. 4f, the photocurrent of the FAPbBr$_3$ photoanode decreased by approximately 25% under UV-filtered light, reducing to about 7.0 mA cm$^{-2}$ at 1.23 V vs. RHE. When the cut-off absorption edge is set at 560 nm, the energy of UV light in AM 1.5 G spectrum accounted for 13%, delivering a maximum photocurrent density of 1.36 mA cm$^{-2}$ (Fig. 4g). When a visible light filter was applied, the photocurrent of FAPbBr$_3$ photoanode is limited to 1.23 mA cm$^{-2}$ at 1.23 V vs. RHE, and the onset potential dramatically increased under UV light. This observation could be explained by the fact that UV light cannot penetrate deep into the perovskite film, leading to the insufficient driving force for the carriers to flow to the HTM side. The stability measurements with filtered light are presented in Fig. 4b and Supplementary Fig. 33. It was found that the FAPbBr$_3$ photoanode degraded

**Table 1 | Stability of the reported perovskite photoanodes for water oxidation**

| Perovskite composition | Year | Stability data (Operation time/residual current) | Ref. |
|---|---|---|---|
| MAPbI$_3$ | 2016 | 0.5 h/64% | 54 |
| MAPbI$_3$ | 2018 | 6 h/0% | 55 |
| (5-AVA)$_x$(MA)$_{1-x}$PbI$_3$[a] | 2019 | 12 h/69% | 56 |
| FA$_{0.80}$MA$_{0.15}$Cs$_{0.05}$PbI$_{2.55}$Br$_{0.45}$ | 2021 | 13 h/84% | 57 |
| CsPbBr$_3$[a] | 2019 | 35 h/7% | 13 |
| CsPbBr$_3$[a] | 2023 | 24 h/78% 70 h/77% | 58 |
| FA$_{0.83}$Cs$_{0.17}$Pb(I$_{0.8}$Br$_{0.2}$)$_3$ | 2021 | 47 h/0% | 59 |
| FA$_{0.93}$MA$_{0.07}$PbI$_3$ | 2022 | 12 h/86% 48 h/50% | 60 |
| FAPbBr$_3$[a] | 2023 | 12 h/100% 24 h/94% 60 h/43% | This work |
| FAPbBr$_3$[a] (UV filter)[b] | 2023 | 100 h/95% 125 h/57% | This work |

[a]HTM-free device.
[b]After 72 h of operation, the device rested in the dark for 24 h before further testing.

rapidly under UV light, with a current half-life of only 2.2 h, confirming the sensitivity of the perovskite photoanode system to UV light. The color of the FAPbBr$_3$ layer also turns white, indicating the rapid degradation of the ITO-SnO$_2$/FAPbBr$_3$ interface. In contrast, under UV-free illumination, the photoanode is much more stable under operating conditions for 50 h, retaining almost 100% of the initial photocurrent and gradually decaying to 88% of the initial photocurrent for an additional 24 h. More importantly, the photocurrent can be restored to the initial level after resting in the dark for 12 h at open circuit conditions, as commonly observed in regularly structured perovskite solar cells with n-i-p architectures[31–34]. The reversible losses and dark recovery phenomenon are typically attributed to ion and defect migration within the perovskite layer and at the interfaces between perovskite and selective contacts[32]. The system demonstrates remarkable operational stability, with no current decay even after cumulative 96 h of electrolysis. Eventually, the photocurrent density retains more than 57% of its initial level even after 125 h. As revealed in Supplementary Fig. 34 and the inset image in Fig. 4b, the current reduction should be due to a sealing failure in the electrolyte, which led to moisture penetration on the edges. Nevertheless, the center of the perovskite layer does not turn white, indicating its intrinsic stability toward water oxidation catalysis under visible light, which also implies that further improvements in sealing materials and processes can further enhance device stability. To the best of our knowledge, the as-fabricated FAPbBr$_3$ photoanode is the first lead halide perovskite-based photoanode with a stability of more than 100 h to date (Table 1). Although the use of a UV filter would cause an additional performance penalty, the saturation current of 7 to 8 mA cm$^{-2}$ is still superior to oxide-based photoanodes like BiVO$_4$ and WO$_3$, making FAPbBr$_3$ perovskite a competitive candidate for low-cost photoanode materials.

### Photothermal effect of photoanode
While it has been demonstrated that increased electrolyte temperatures can enhance the PEC performance of oxide-based photoanodes[35,36], the effect of temperature on perovskite photoanode performance remains uncertain. Introducing the photothermal effect into the catalytic process has an evident advantage, as increased temperature can accelerate the kinetics of electrochemical reactions. However, in the view of charge separation, higher temperatures are detrimental to the performance and stability of perovskite solar cells[37,38], which presents a challenge for balancing these two competing factors. Therefore, investigating the photothermal effects on perovskite-based photoanodes is crucial. The question to be

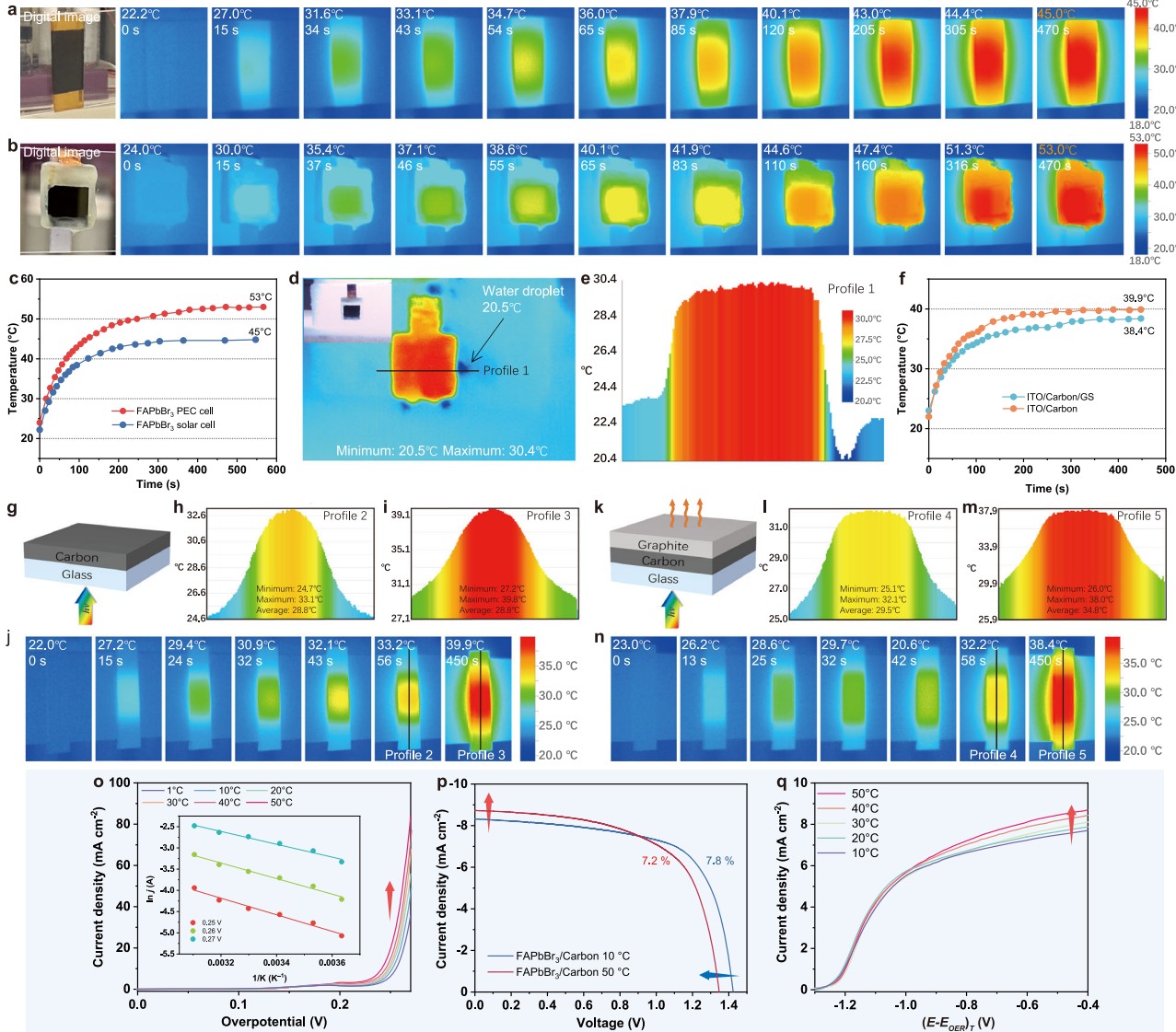

**Fig. 5 | Photothermal performance of FAPbBr₃ photoanode.** Thermal images of (**a**) FAPbBr₃/carbon solar cell and (**b**) FAPbBr₃ photoanode under continuous light illumination (AM 1.5 G, 100 mW cm⁻²). **c** Corresponding heating curves of FAPbBr₃/Carbon solar cell and FAPbBr₃ photoanode. **d** Thermal image and (**e**) corresponding temperature profile of FAPbBr₃ photoanode heated by continuous light illumination in the electrolyte (AM 1.5 G, 100 mW cm⁻²). **f** Heating curves of ITO/carbon and ITO/carbon/GS under continuous light illumination (AM 1.5 G, 100 mW cm⁻²). **g** Structure diagram of carbon-coated ITO glass. **h**, **i** Temperature profiles and (**j**) thermal images of ITO/carbon film under continuous light illumination (AM 1.5 G, 100 mW cm⁻²). **k** Structure diagram of GS/carbon-coated ITO glass. **l**, **m** Temperature profiles and (**n**) thermal images of ITO/carbon/GS film under continuous light illumination (AM 1.5 G, 100 mW cm⁻²). **o** Temperature-dependent LSV curves of GS/NiFe alloy/NiFe LDH catalyzed OER in 1.0 M KOH (scan rate: 5 mV s⁻¹). Inset: Arrhenius plots under different overpotentials (**p**) Temperature-dependent density-voltage (*J-V*) curves of FAPbBr₃/Carbon solar cell under light illumination (AM 1.5 G, 100 mW cm⁻²). **q** Temperature-dependent LSV curves of FAPbBr₃ photoanode under light illumination (scan rate: 50 mV s⁻¹, AM 1.5 G, 100 mW cm⁻²).

addressed in this section is whether high temperatures have a positive or negative impact on photoanode performance.

Prior to examining the catalytic performance, the photothermal characteristics of HTM-free FAPbBr₃ solar cell and photoanode were evaluated by recording the time-dependent temperature variation in ambient air (Supplementary Fig. 35). As shown in Fig. 5c, when FAPbBr₃ solar cell is exposed to simulated AM 1.5 G light (100 mW cm⁻²), the temperature rises by 14 °C within 60 s and reaches 45 °C within 470 s, indicating a pronounced photothermal effect. The infrared image presented in Fig. 5a implies that the heated region is primarily concentrated in the central area covered by the carbon layer, indicating that the carbon conduction layer can serve as a photothermal layer. Moreover, the photothermal performance of the FAPbBr₃ photoanode is enhanced after encapsulation with a

graphite sheet and epoxy resin, as the temperature immediately rises upon illumination and the temperature increases at 470 s are found to be 30 °C (Fig. 5c). As shown in Fig. 5b, the heat generated by the photoanode is primarily concentrated in its central region, while the surrounding epoxy resin functions as a competent insulator, inhibiting heat dissipation. The photothermal effect of the FAPbBr₃ photoanode in electrolyte was also assessed using the infrared camera. As presented in Fig. 5d, the surface temperature of the photoanode reaches over 30 °C after 300 s continuous illumination in 1.0 M KOH, implying that light can heat the interior of the photoanode even in the electrolyte. The temperature profile reveals that the photoanode surface is evenly heated, with a temperature differential of 10 °C between the photoanode surface and the electrolyte (Fig. 5e).

Carbon and carbon/graphite layers were coated on the ITO glass to investigate the photothermal conversion properties of carbon and graphite components separately (the structures are shown in Figs. 5g and 5k). Under AM 1.5 G irradiation, the ITO/carbon film exhibits a considerable photothermal conversion, with an 18 °C temperature increase after 300 s of exposure. Conversely, after attaching the graphite sheet, the temperature rises of the ITO/carbon/GS film decreased by 1.5 °C compared to that of the ITO/carbon film (Fig. 5f). This decrease in temperature could be attributed to the enhanced surface heat dissipation in the carbon/GS film, as graphite sheet are widely employed as thermal pads in diverse industries due to their superior heat dissipation characteristics. Fig. 5j and Fig. 5n show the thermal images of the ITO/carbon and ITO/carbon/GS films, respectively, under continuous light illumination. The graphite sheet, which exhibits excellent thermal conductivity, shows a uniform temperature distribution. As illustrated in Fig. 5h, i, the heat primarily accumulates in the center of the ITO/carbon film, producing parabolic temperature profile curves. In contrast, ITO/carbon/GS film exhibits uniform heat distribution on the surface, resulting in trapezoidal-shaped temperature profile curves (Fig. 5l, m). Based on this analysis, it can be concluded that the carbon layer primarily generates the photothermal effect in the device; moreover, the graphite layer utilized for conductive sealing also functions as a thermal pad layer, facilitating the rapid and uniform heat conduction toward the catalyst layer to mitigate the accumulated heat damage to the perovskite layer.

The use of a GS separator also allows for the individual investigation of the temperature effect on electrochemical water oxidation catalysis of the ITO/GS/NiFe alloy/NiFe LDH electrode. Temperature ($T$)-controlled electrocatalysis was conducted in 1.0 M KOH to determine the temperature dependency for the electrode. In an electrochemical reaction, the activation overpotential ($\eta$) is the potential difference from the equilibrium value required to produce a current, which depends on the activation energy of the redox event. Hence, the catalytic rate, as indicated by the current densities, should be compared at the same overpotential in order to accurately assess the catalytic activity (see Supplementary Note 5 for the calculation of overpotential). After calibration with driven force, the corresponding Arrhenius plots of the current at various potentials against $1/T$ are calculated and shown in the inset figure of Fig. 5o. At an overpotential of 250 mV, the apparent activation energy is calculated to be 16.2 kJ mol⁻¹. The activation energy at zero overpotential is derived from the intercept of apparent activation energy vs. overpotential plots at $\eta = 0$ (Supplementary Fig. 38)[39]. The obtained value (65 kJ mol⁻¹) is smaller than that of other nickel-based catalysts, such as NiTe (100.2 kJ mol⁻¹) > Ni₂Si (98.6 kJ mol⁻¹) > Ni (94.0 kJ mol⁻¹) > NiAs (86.3 kJ mol⁻¹) > NiP (78.4 kJ mol⁻¹)[40], indicating faster intrinsic OER kinetics for the electrodeposited NiFe alloy/NiFe LDH catalysts.

The temperature effect on the FAPbBr₃ solar cell is studied using a lab-made setup (Supplementary Fig. 39). As shown in Fig. 5p, an increase in temperature will cause a decrease in the $E_g$ of the semiconductor material, resulting in a slight increase in the short-circuit current[38]. However, this increase in $J_{sc}$ is not enough to compensate for the loss of FF and $V_{oc}$ values due to higher recombination probabilities, which ultimately leads to a reduced PCE at higher temperatures[41]. On one hand, for photoanode devices with perovskite-based light-absorbing layer, the decrease in voltage is certainly detrimental to the performance of photoanode devices, particularly in terms of onset potential and maximum ABPE. On the other hand, the reduction in $E_g$, as well as the increase in $J_{sc}$, can lead to a higher saturated current under higher applied potential. Additionally, an increase in temperature can significantly decrease the theoretical potential of the water oxidation reaction while greatly increasing the catalytic current under the same driven force. These competing effects make it uncertain how temperature affects the performance of photoanode devices, and thus, it is necessary to study the temperature effect on complete

perovskite photoanodes. Fig. 5q shows the photocurrent density-overpotential curves of FAPbBr₃ photoanode at different temperatures. The results indicate that changes in temperature do not significantly affect the onset potential under the same driving force for the reaction, which suggests that the decrease in voltage for light-absorbing perovskite material and the reduction in overpotential requirements for the catalytic process offset each other, resulting in a consistent apparent onset potential for FAPbBr₃ perovskite photoanode. Furthermore, the increased short-circuit current in the light-absorbing portion and the increased catalytic current in the catalytic portion leads to a significant improvement in the saturated catalytic current of the photoanode. Based on the above analysis, it can be concluded that the increase in temperature has a positive effect on the photoanode performance. Utilizing photothermal effect can be an effective strategy to further enhance the performance of perovskite-based photoanodes, provided that stability is not compromised. Moreover, the FAPbBr₃ material exhibits superior thermostability compared to other candidates, such as MAPbBr₃ and MAPbI₃, making it an ideal building block for highly efficient photoanodes that can fully leverage the photothermal effect without suffering from thermal degradation.

Solar hydrogen can be produced using a variety of methods, including the use of PV cells, PEC cells, and hybrid systems that combine both technologies. Our work demonstrates that PVM-PEC configuration can fully utilize the photothermal effect, like PEC, to maximize solar energy utilization (see summary in Supplementary Note 6). As-fabricated FAPbBr₃ photoanode can be synthesized with remarkable accessibility without requiring high temperature, inert atmosphere, or vacuum evaporation techniques. The carbon/GS protection strategy effectively prevents water corrosion from damaging the sensitive perovskite, and the photothermal properties of the carbon material allow for extended spectral utilization. Meanwhile, this HTM-free and precious metal-free structure dramatically reduces fabrication costs. The scalable electrodeposition preparation of catalyst layers enables facile fine-tuning of the composition, structure, and morphology of electrocatalysts using electrochemical technologies. We believe that these findings have significant implications for the development of cost-effective, high-performance integrated photoelectrodes for solar energy conversion applications.

## Discussion

In summary, this work has demonstrated a FAPbBr₃-based photoanode featuring a photocurrent density around 8.5 mA cm⁻² at 1.23 V vs. RHE, an onset potential below 0 V vs. RHE, and half-life stability over 55 h using a FAPbBr₃ absorber, a carbon/graphite conductive protection layer and NiFe alloy/NiFe LDH catalysts under alkaline conditions. Due to its fast catalytic kinetics, high photovoltage, and rational band structure, a high ABPE of 8.5% was reached at 0.08 V vs. RHE. Further stability studies and post-characterizations revealed that the perovskite interface was damaged by UV light, resulting in a detached absorber layer and performance loss. Accordingly, by applying a UV filter, the device achieved record stability of over 100 h in 1.0 M KOH under constant simulated solar illumination, with currents around 7.5 mA cm⁻² at 1.23 V vs. RHE. More importantly, this work has conducted a detailed investigation of the photothermal effect on FAPbBr₃ photoanode, confirming that the photoanode based on photovoltaic materials can also take full advantage of the photothermal effect and improve catalytic performance. In the design structure of ITO/SnO₂/FAPbBr₃/carbon/GS/NiFe alloy/NiFe LDH, the carbon layer serves as both the hole transport layer and the photothermal layer, while the graphite layer functions not only as a waterproof and conductive layer but also efficiently transfers heat, serving as a thermal homogenization layer. Meanwhile, the dense electrodeposited alloy layer provides a robust substrate for high-performance LDH catalyst loading, ensuring efficient electrical and thermal conductivity. The compact and

multifunctional design endows the monolithic device with high performance and excellent stability. These observations, along with the ultralow onset potential and the facile processability of FAPbBr$_3$ photoanode, demonstrate the potential of lead halide perovskite-based PEC as a promising approach to achieving efficient, cost-effective, and scalable photoelectrochemical solar fuel production.

## Methods

### Chemicals

Colloidal tin oxide solution (SnO$_2$, 15% in H$_2$O) was purchased from Alfa Aesar. Sodium hydroxide (semiconductor grade, 99.99%), potassium hydroxide (semiconductor grade, 99.99%), tri-Sodium citrate dihydrate (99%), Ammonium sulfate (99.0%), sodium hypophosphite monohydrate (99.0%), Nickel(II) sulfate hexahydrate (99.99%), Iron(II) sulfate heptahydrate (99.0%), Nickel(II) chloride hexahydrate (99.9%), Iron(III) chloride (anhydrous, 99.99%), Nickel(II) nitrate hexahydrate (99.999%), Iron(III) nitrate nonahydrate (99.95%), dimethylformamide (DMF, anhydrous, 99.8%), dimethyl sulfoxide (DMSO, anhydrous, 99.9%), acetonitrile (HPLC), chlorobenzene (anhydrous, 99.8%) methanol (HPLC, 99.9%), bis(trifluoromethanesulfonyl)imide lithium (Li-TFSI) and 4-tert-butylpyridine (TBP) were purchased from Sigma-Aldrich and used as received without further purification. 2,20,7,70-tetrakis [N, N-di(4-methoxyphenyl) amino]−9,90-spirobifluorene (SpiroOMeTAD) was purchased from Borun. PbBr$_2$ (99.99%, trace metals basis) was purchased from TCI Chemicals. FK209 [tris(2-(1Hpyrazol-1-yl)−4-tert-butylpyridine)-cobalt (III) tris(bis(trifluoromethylsulfonyl)imide) and FABr were provided by Greatcell solar. Carbon pastes (DN-CP01) were purchased from Dyenamo. 160 um graphite thermal sheet (794−3973) was purchased from RS components. High-purity deionized water (18.2 MΩ cm$^{-1}$), supplied with a Milli-Q system (Millipore, Advantage A10), was used in all experiments. Indium tin oxide (ITO) glass substrate and epoxy adhesive (Loctite 9466) were purchased from a local company. All other reagents were commercially available and used as received. Organic solvents were of analytical reagent grade and used without further purification.

### Fabrication of FAPbBr$_3$ solar cell

The structure of hole transport materials-free FAPbBr$_3$ solar cell is glass/ITO/SnO$_2$/FAPbBr$_3$/carbon. ITO glass was cleaned with detergent, acetone, isopropanol, and water for 20 min, respectively. UV/ozone (UVO) treatment was then applied for 30 min before use. SnO$_2$ film was deposited on the ITO substrate by spin-coating method using the commercial SnO$_2$ colloidal solution (diluted with deionized water, the volume ratio of water to SnO$_2$ is 4:1) at 3000 rpm for 30 s, followed by annealing treatment at 150 °C on a hotplate for 30 min. After cooling down, UV/ozone treatment was conducted on SnO$_2$ film for 20 min before further use. The FAPbBr$_3$ film was prepared using a two-step spin coating method. PbBr$_2$ precursor solution was prepared by dissolving 1.3 M of PbBr$_2$ in a mixture of DMF and DMSO with a volume ratio of 9:1 and stirred at 70 °C overnight. Then the PbBr$_2$ precursor solution was spin-coated onto SnO$_2$ substrates at 3000 rpm for 30 s, followed by annealing treatment in the ambient conditions at 70 °C for 1 min. Next, 0.065 g FABr was dissolved in 1.0 ml methanol and spin-coated on top of PbBr$_2$ precursor film (3000 rpm, 30 s), followed by annealing treatment in the ambient conditions at 150 °C for 25 min. Finally, A carbon paste (DN-CP01) was doctor-bladed as the top contact on FAPbBr$_3$ film. Devices were further annealed at 100 °C for 30 min in the air for the formation of a solid carbon layer. The reference solar cell with structure of glass/ITO/SnO$_2$/FAPbBr$_3$/spiro-MeOTAD/Au was fabricated using same procedure. The HTM layer was prepared by spin coating method: 70 mM Spiro-OMeTAD was dissolved in chlorobenzene with additives of TBP, Li-TFS (1.8 M in acetonitrile), and FK209 (0.25 M in acetonitrile). The Molar ratio of Spiro-OMeTAD: TBP: Li-TFSI: FK209 was 1: 3.3: 0.5: 0.05[42]. The Spiro-OMeTAD solution was spin-coated on

the perovskite films (4000 rpm, 30 s) and 80 nm Au layer was then deposited via vacuum evaporation on top of spiro-OMeTAD as a contact electrode. The devices were stored in a dry box overnight before the J-V measurement.

### Fabrication of NiFe catalysts

The structure of catalyst layers is glass/ITO/GS/NiFe alloy/NiFe LDH. The self-adhesive GS layer was stuck onto the ITO glass. The NiFe alloy layer was electrodeposited on the GS layer (1.0 × 1.0 cm) using a reported method with subtle modification[43]. In brief, sodium citrate dihydrate (20.0 mmol, 6.00 g), ammonium sulfate (45.0 mmol, 6.00 g), and sodium hypophosphite monohydrate (57.0 mmol, 6.00 g) were dissolved in 100 mL water at ambient temperature. Nickel sulfate hexahydrate (10.0 mmol, 2.62 g) and iron(II) sulfate heptahydrate (2.0 mmol, 0.54 g) were added into the solution with stirring. The pH of the solution was adjusted to 10.0 using 10 M NaOH solution. The deposition process was performed using a three-electrode electrochemical cell (platinum mesh as the counter electrode and an Ag/AgCl (saturated KCl) as the reference electrode) filled with 15 mL electrodeposition solution at an applied potential of −1.10 V vs. Ag/AgCl for 1600 s. The alloy film was rinsed with deionized water and then dried under an N$_2$ atmosphere before further use. The NiFe LDH layer was electrodeposited on the alloy layer using a modified reported method[44]. The electrolyte bath contained 5 mM nickel(II) nitrate hexahydrate and 5 mM iron(III) nitrate nonahydrate. The electrodeposition was then carried out at −1.0 V vs. Ag/AgCl for 900 s. NiFe LDH was also electrodeposited onto the GS electrode following the same procedures. The LDH film was rinsed with deionized water and dried under an N$_2$ atmosphere before further use.

### Fabrication of FAPbBr$_3$ photoanode

Following Poli's methods[13], after the FAPbBr$_3$ solar cell part was prepared, the above-mentioned 160 um self-adhesive GS layer was stuck onto the carbon film (step 2, Supplementary Fig. 1). The bottom edge part of the attached GS was further covered by carbon paste to ensure conductivity. Then copper tape and copper wire were stuck on the ITO and GS parts, respectively (step 3, Supplementary Fig. 1). Latterly, the photoanode was sealed by chemical-resistant epoxy resin Loctite 9466 (step 4, Supplementary Fig. 1) and hardened at 60 °C for 30 min. The devices were stored in a dry box overnight before subsequent use. The catalyst layers were deposited on GS using the same procedure and conditions as above. In particular, the deposition process was performed using a three-electrode electrochemical cell with the working electrode connected to copper wire. After the preparation is complete, the copper wire should be cut short to prevent possible corrosion in the electrolyte.

### Electrochemical measurements

All electrochemical characterizations were carried out in a standard three-electrode cell connected to a CHI 660E electrochemical workstation, using the prepared films as the working electrode, a Pt mesh as the counter electrode, and Hg/HgO (1.0 M KOH) as the reference electrode. The temperature was controlled at 25 °C by a thermostatic water bath unless otherwise noted. All measured potentials were converted to values relative to RHE by a commercialized RHE reference electrode (HydroFlex of Gaskatel) at 25 °C[45]. All linear sweep voltammetry curves were measured in a three-electrode configuration with iR correction (95%) unless otherwise noted. Chronopotentiometry was recorded under the same experimental setup with iR-correction. All the correction was done manually according to the equation $E_{corr} = E_{meas} − iR_u$, where $E_{corr}$ is iR-corrected potential, $E_{meas}$ is the experimentally measured potential, and $R_u$ is the equivalent series resistance extracted from Nyquist plots (-2.3 Ω in 1 M KOH). Tafel slopes were calculated based on polarization curves by fitting to the

following equation $\eta = b \times \log(j) + a$, where $j$ is current density (mA cm$^{-2}$), $\eta$ is overpotential (mV), and b is Tafel slope.

## Photoelectrochemical measurements

All photoelectrochemical measurements were carried out at room temperature by using a CHI 660E electrochemical workstation. The photoelectrochemical performances of photoanodes were measured with a three-electrode configuration, using the prepared photoanodes as the working electrode, a Pt mesh as the counter electrode, and Hg/HgO (1.0 M KOH) as the reference electrode. All measured potentials were converted to values relative to RHE by a commercialized RHE reference electrode (HydroFlex of Gaskatel) at 25 °C. The simulated solar illumination was obtained by a NEWPORT LCS-100 solar simulator (type 94011A-ES, a 100 W Xenon arc lamp with an AM 1.5 G filter). The incident light power intensity was calibrated to 100 mW cm$^{-2}$ by a Thorlabs S401C photometer (190 nm −20 μm). $J$-$V$ curves were obtained by LSV without $iR$-correction. The chopped light was controlled using an electronic shutter and corresponding driver (type 76995) provided by Newport. The long-term electrolysis was carried out with a constant potential under the same experimental setup. The ABPE was calculated from the $J$-$V$ curves (in a three-electrode system) under illumination according to the following equation:

$$\text{ABPE}(\%) = \frac{(1.23 - V_{RHE}) \times (j_{light} - j_{dark})}{P_{light}} \times 100\% \qquad (1)$$

where $P_{light}$ was the light intensity of incident illumination, $V_{RHE}$ indicated the applied potential in RHE scale, $j_{light}$ and $j_{dark}$ were photocurrents and dark current, respectively. The IPCE values of photoanodes were calculated using the following equation:

$$\text{IPCE}(\%) = \frac{1240 \times (j_{light} - j_{dark})}{\lambda \times P_\lambda} \times 100\% \qquad (2)$$

The monochromatic light was produced using a monochromator. The light intensity ($P_\lambda$) at each wavelength ($\lambda$) was determined by a photometer; $j_{light}$ and $j_{dark}$ were photocurrents and dark current, respectively, measured by an electrochemical workstation. The onset potential ($E_{onset}$) for photoanode was defined as the intersection point of the $J$-$V$ curve with the x-axis. The corresponding photovoltage ($V_{ph}$) was calculated according to the following equation:

$$V_{ph} = 1.23 - E_{onset}(25°C) \qquad (3)$$

It's worth noting that this approach neglects catalytic overpotentials, resulting in a potential underestimation of the photovoltage for PEC devices.

## Computational details

All the DFT calculations were performed using the Vienna Ab-initio Simulation Package (VASP)[46]. The projector-augmented wave (PAW) pseudopotentials method[46,47] was utilized to describe the electron-ion interactions with the plane-wave basis expansion cut-off energy set to be 500 eV. The generalized gradient approximation (GGA) was used with Perdew–Burke–Ernzerhof (PBE)[48] to perform the exchange-correlation potential. The valence states of the potential chose $5d^{10}6s^2 6p^2$ for the lead (Pb) atom, $4s^2 4p^5$ for the bromine (Br) atom, $2s^2 2p^2$ for the carbon (C) atom, $2s^2 2p^3$ for the nitrogen (N) atom, and $1s^1$ for the hydrogen (H) atom. The Hubbard-type correction was set on Pb $6p$ and Br $4p$ orbitals with a U-J value of 9 eV and 8 eV, respectively[49,50]. The $p(2 \times 2 \times 2)$ supercell of FAPbBr$_3$, consisting of 96 atoms, is considered for the defect study. A $5 \times 5 \times 5$ Monkhorst–Pack k-point mesh sampling for Brillouin-zone integration was used for the bulk optimization, while it was $2 \times 2 \times 2$ for all defective structures. The equilibrium

was reached when the forces on the relaxed atoms became less than 0.02 eV/Å. The van der Waal (vdW) interaction was described by the DFT-D3 method[51,52]. Note that, due to the presence of heavy atoms, spin-orbit coupling (SOC) is also included through all calculations[53].

## Data availability

The complete dataset in the main text is available at http://zenodo.org with the identifier https://doi.org/10.5281/zenodo.8242075. Other data that support the findings of this study are available from the corresponding author upon reasonable request.

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

## Acknowledgements

L.S. acknowledges the financial support from the National Key R&D Program of China (2022YFA0911902), the National Natural Science Foundation of China (Grants No. 22088102) and the starting-up package of Westlake University. Y.L. and E.M.J.J. acknowledge the financial support obtained from the Swedish Energy Agency, ÅForsk, Swedish Research Council (VR), and Olle Engkvist Foundation. G.B. thanks support from the STandUP for Energy program. F.L. acknowledges the financial support from the National Natural Science Foundation of China (Grants No. 22172011). The authors acknowledge Westlake University High-Performance Computing Center for computation support. The authors thank the Instrumentation and Service Center for Physical Sciences at Westlake University for facility support and technical assistance.

## Author contributions

H.Y., Y.L., E.M.J.J., and L.S. conceived the project design and initiated the project. H.Y. and Y.L. performed synthesis, structural and electrochemical characterizations. Y.D. performed DFT calculations. H.Y., Y.L., F.L., L.W., B.C., F.Z., T.L., and G.B. contributed to data analysis and interpretation. All authors contributed to discussions. H.Y. and Y.L. wrote the original draft with inputs from the other authors, and all authors reviewed and revised the manuscript. E.M.J.J. and L.S. supervised the research.

## Funding

## Competing interests

The authors declare no competing interests.
