## [Peer Review File · Nature Communications]

Monolithic FAPbBr₃ Photoanode for Photoelectrochemical Water Oxidation with Low Onset-potential and Enhanced StabilityREVIEWER COMMENTS

Reviewer #1 (Remarks to the Author):

The manuscript reports monolithic FAPbBr₃ photoanode with low onset potential and >100 h stability in water splitting driving condition without UV light. Although a low onset potential close to 0V was achieved in a perovskite based photoanode, it doesn't seem to exhibit a stable performance even when deprived of UV light. Nevertheless, there is significance in providing a detailed explanation of the photothermal effect and the low onset potential, which were not previously addressed. The data and explanations provided seem to be sufficient in this regard. These are following questions to be addressed for authentic publication.

Comments 1: Generally, in perovskite solar cell, the incorporation of spiro-oMeTAD as the hole transport layer, along with the use of gold electrode, can improve charge extraction and reduce recombination losses, leading to enhanced overall device performance compared to HTL-free carbon cell. How can an HTL-free carbon cell be more efficient than the spiro/Au cell?

Comments 2: In FAPbBr₃ photoanode, the stability of photoanode appears to be more influenced by the stability of the perovskite solar cell than other factors, so it seems important to understand the characteristics of this aspect. In Figure S2 and S7, the HTM-free FAPbBr₃/Carbon solar cell showed improved efficiency and reduced hysteresis after aging under dry air conditions. What could be the cause of this, and did these characteristics also affect the stability of the photoanode?

Comments 3: From the same perspective as the previous question, what is the stability measurement method under one-sun condition (MPPT or open circuit condition, etc.)? Also, during the initial stage, the efficiency tends to increase. What could be the cause of this, and did these characteristics also affect the stability of the photoanode?

Comments 4: Graphite sheets are not typically used as a standalone material for preventing for penetration. In Figure 4, the form of degradation appears to be due to water penetration. Even if electrolyte is placed on top of a graphite sheet without operating photoanode, wouldn't it prevent water penetration?

Comments 5: In line 180~182 and Figure 2e, the ABPE value isn't best of those reported for photoanodes, it should be revised.

Comments 6: It was known that the adhesion between SnO₂ layer and FAPbBr₃ was strong. And authors fabricate FAPbBr₃ photoanode with high adhesive epoxy resin. How the tested sample could be easily separated into two parts?

Comments 7: In Figure 6 and the corresponding manuscript, authors defined the FAPbBr₃ photoanode as a PVPEC configuration and explained its advantages. However, the PVPEC system refers to a combination of a photovoltaic (PV) cell and a photoelectrochemical cell (PEC) [1,2], and the FAPbBr₃ photoanode does not consist of a combination of two cells that absorb light; it only has a single light-absorbing layer(perovskite). Therefore, it appears that a different expression should be used to describe it.

[1] M. Bake et al. Metal Halide Perovskites for Solar-to-Chemical Fuel Conversion. *Adv. Energy Mater.* 10, 1902433 (2020).

[2] Park et al. Water Splitting Progress in Tandem Devices: Moving Photolysis beyond Electrolysis. *Adv. Energy Mater.* 6, 1600602 (2016).

Reviewer #2 (Remarks to the Author):

The authors have reported a metal-halide perovskite-based photoanode for PEC water oxidations and investigated its physical mechanism using experimental and DFT methods. They proposed the potential of lead halide perovskite-based PEC as a promising approach to achieving efficient, cost-effective, and scalable PEC solar fuel production. Using the DFT calculations, the author suggested that the vacancy structure of Br leads to an n-type property of the FAPbBr₃ film. However, the manuscript needs further revisions to clarify the following points:

- (1) PbBr can shift the Fermi level penetrating into CBM, why authors suggested that only the vacancy structure of Br leads to an n-type property of the FAPbBr₃ film?
- (2) Does the author calculate the defect formation energy to determine the type of defect that can be formed in the metal-halide perovskite? Whether the Br vacancy has the lowest defect formation energy?
- (3) The author performed spin-polarized DFT calculations. But the density of states (DOS) are not spin polarization. Please provide the spin-polarized DOS.
- (4) Fig. S21 does not label the Fermi level.

Reviewer #3 (Remarks to the Author):

This article presents FAPbBr₃ halide perovskite photoanodes for PEC water oxidation. Halide perovskites are known to be excellent semiconductors for solar cells, but because they are unstable in water their utilisation in other fields such as aqueous photoelectrochemistry is limited. The authors use a previous method published in Nature Comm 2019 by Poli et al. to protect the solar cell and be able to use it as a photoanode. There are some strengths in the article such as a low onset potential and investigation of expected photothermal effects, as well as systematic characterisation and explanations that make the article enjoyable to read and useful. There are also weaknesses, that I list below as follows:

- The word limit of 5000 words is exceeded in 1000 words
- This article follows Poli's method presented in Nature Comm 2019, 2097, where mesoporous carbon layer, adhesive graphite sheet and electrocatalyst were used to protect and use a bromide perovskite for PEC water oxidation. However, the introduction or the Methods omits this relevant citation. Poli's article is cited, but only at the end in a comparison of performances. Poli's article should have been cited early on, otherwise it may be seen as appropriating other people's work
- The ABPE% measurement appears not correct. ABPE values should be measured in 2-electrode configuration. Measurements in 3-electrode configuration are not ABPE. Please see for example: Nat Energy 2, 16191 (2017) <https://doi.org/10.1038/nenergy.2016.191> or textbooks of PEC
- The low onset potential is great but is practically attributed to the quality of the semiconductor FAPbBr₃, which is known. Such quality is necessary but not sufficient. Unlike solar cells, PEC devices performance strongly depend on the catalytic performance. If there is no catalysis, there is no photocurrent. The good translation from solar cell photocurrents to PEC photocurrents rely on the quality of the method to protect and extract the charges to the electrolyte, and this is achieved because Poli's method of using mesoporous carbon and graphite sheets functionalised with electrocatalyst is used. This is not properly conveyed, emphasized, and studied in the article to explain the good onset potential
- The use of bromide perovskite, mesoporous carbon and graphite sheet with a NiFe-based electrocatalyst and >100 h stability has already been presented in conferences and literature (DOI: 10.26434/chemrxiv-2023-cl4m5) and should be cited
- The introduction generates expectations on bias-free PEC devices but these are later not presented in the article. The measurements are always in 3-electrode configuration
- The introduction also generated expectations about the use of the solar broad spectrum, but later the UV part of the solar spectrum can't be used because it degrades the organic part of the perovskite.

- Page 4: sentence "Additionally, its inorganic counterparts require high-temperature annealing over 300 °C for stabilization, which leads to poor crystallinity in the films.¹²" This needs rewording. It sounds like if crystallinity decreases with temperature, which is the other way around.
- Figure 1d shows that the VB edge of FAPbBr₃ is not suitable for water oxidation, contrary to what is shown later in Figure 3. The difference may be in the different pH values, but this may confuse readers if not properly explained
- Sentences "In our study, the theoretical oxidation potential of Br⁻ is 1.087 V vs. NHE (pH = 0), which is equivalent to -5.59 eV vs. vacuum level.³⁰ Under visible light irradiation, the potential is almost the same as the VBM of perovskite. Thus, self-oxidation is less likely to occur on FAPbBr₃ because the reaction requires a certain additional driving force." There are different things here that are not clear. An oxidation happening at 1.087 V vs. NHE (pH = 0) is said not likely, but water oxidation at 1.23 V vs NHE pH 0 (Fig 1) is reported. Moreover, it is not clear what self-oxidation they are referring to.
- Table 1, CsPbBr₃ PEC devices are claimed to last 17h but they lasted 35h in ref. 27
- The use of lower and higher CB and VB edge is confusing. Shallower and deeper would be less confusing

REVIEWER COMMENTS

Reviewer #1 (Remarks to the Author):

The manuscript reports monolithic FAPbBr₃ photoanode with low onset potential and >100 h stability in water splitting driving condition without UV light. Although a low onset potential close to 0V was achieved in a perovskite based photoanode, it doesn't seem to exhibit a stable performance even when deprived of UV light. Nevertheless, there is significance in providing a detailed explanation of the photothermal effect and the low onset potential, which were not previously addressed. The data and explanations provided seem to be sufficient in this regard. These are following questions to be addressed for authentic publication.

Response: We're grateful to the reviewer for their insightful feedback on our manuscript. The constructive comments were valuable in refining our work. We have carefully considered your points and revised the manuscript accordingly to address all concerns and questions.

Comments 1: Generally, in perovskite solar cell, the incorporation of spiro-oMeTAD as the hole transport layer, along with the use of gold electrode, can improve charge extraction and reduce recombination losses, leading to enhanced overall device performance compared to HTL-free carbon cell. How can an HTL-free carbon cell be more efficient than the spiro/Au cell?

Response: We thank the reviewer for the comment. Indeed, the highest efficiency achieved in the perovskite solar cells is usually reported based on the "spiro-OMeTAD/Au" (*Nature*, 377, 6605, 531 (2021); *Science*, 377, 6605 (2022)). However, unlike our ambient condition fabrication processes, perovskite solar cells are normally fabricated under a moisture-controlled environment in those reports. As it is well known that perovskites are very moisture-sensitive due to their ionic components, pinholes are usually formed in the perovskite film if no moisture control has been made. Spiro-OMeTAD is a kind of small organic molecule, which has a chance to penetrate these pinholes and contact with the SnO₂ bottom layer. The direct contact of hole transport material with electron transport material could increase the dark current and deteriorate the photovoltaic performances. On the other hand, the conductivity of pristine spiro-OMeTAD is very low, less than 10E⁻⁵ S cm⁻¹. To improve its conductivity, t-BP and Li-TFSI are usually needed. These additives are highly hydrophilic, which favors the penetration process of the spiro-OMeTAD, and the situation is even worse when the whole device fabrication process is completed in ambient conditions. But carbon paste is consisting of graphite, which possesses a flake shape and big size, see **Fig. S2**. Carbon layer not only can cover these formed pinholes from the surface, but also can protect the perovskite layer due to its hydrophobic property. Consequently, we think that it is very hard to obtain a very high performance based on "spiro-OMeTAD/Au" under ambient conditions, even though spiro-OMeTAD exhibited a very strong hole extraction ability. In fact, the efficiency of our solar cells based on "spiro-OMeTAD/Au" is still comparable to recent reports based on the same device structure (*J. Mater. Chem. A*,

2022,10, 672-681). Furthermore, spiro-OMeTAD does not represent an optimal hole-transporting material for bromide-based perovskites. This is due to the significant energy difference of over 0.6 V between the HOMO of spiro-OMeTAD (-5.0 eV) and the VBM of FAPbBr₃ (-5.6 eV). This discrepancy can result in excessive voltage loss, thus highlighting the importance of developing cost-effective hole-transporting materials that are compatible with bromide-based perovskites. Additionally, it has been observed by us that undoped P3HT exhibits energy levels (-5.24 eV) more closely matched than those of spiro-OMeTAD. Under ambient preparation conditions, an open circuit voltage approaching 1.5 V and an efficiency of 9.4% can be realized (*Fig. 1 for Reviewer #1*). This observation underscores that a spiro-OMeTAD HTL, while suitable for narrow-bandgap perovskites, may not necessarily represent the optimal choice for wide-bandgap perovskites (Of course, this topic extends beyond the scope of the current study and, as such, will not be detailed further in this context). This remains an area worthy of further exploration and research. The superior photovoltaic performance of carbon paste-based device also provides a hint for ambient device fabrication strategy, which can decrease the fabrication cost.

Fig. 1 for Reviewer #1 Performance of FAPbBr₃ solar cells with HTL of P3HT and spiro-OMeTAD.

Comments 2: In FAPbBr₃ photoanode, the stability of photoanode appears to be more influenced by the stability of the perovskite solar cell than other factors, so it seems important to understand the characteristics of this aspect. In Figure S2 and S7, the HTM-free FAPbBr₃/Carbon solar cell showed improved efficiency and reduced hysteresis after aging under dry air conditions. What could be the cause of this, and did these characteristics also affect the stability of the photoanode?

Response: We fully agreed with the reviewer that the stability of the photoanode is significantly influenced by the stability of the absorber layers, i.e., perovskite solar cells.

Perovskite cells using carbon-based structures have been reported to show superior long-term stability due to their good film-forming ability and hydrophobic properties (*Sci Rep* **3**, 3132 (2013), science 345 (6194), 295-298). We find that the PCE improvement of the aged solar cell devices stems from an increase in the short-circuit current (J_{sc}), while there is a decrease in the open-circuit voltage (V_{oc}). This decrease in V_{oc} can be attributed to several factors:

1. The decreased shunt resistance can cause energy loss related to ohmic loss. Usually, the device shunt resistance is also trap-states related.
2. The increase of trap states inside the perovskite absorber and/or charge transport interfaces can improve the chance of non-radiative recombination.
3. The energy level alignment change between the charge transport layer and the perovskite layer, where energy loss happens during the charge extraction process.

The V_{oc} can be heavily influenced by both shallow and deep trap states. Given that the J_{sc} continued to increase during the aging process, it seems unlikely that there was a significant increase in deep-trap states, as these would also seriously impact J_{sc} . Furthermore, alterations in energy level alignment could be another plausible explanation. For instance, it has been reported that SnO_2 , in the electron transport layer, has the potential to react with perovskite due to ion migration, thereby forming a new interface that could shift the interface energy level alignment (*Adv. Mater.* 2022,34, 2110438). Concerning the carbon electrode layer, we used organic polymers (for adhesive reagents) and high-boiling point organic solvents (for graphite dispersion and adhesive polymer dissolution). These organic polymers might gradually interact with the top surface of the perovskite, potentially altering the energy alignment, and thereby impacting the interface charge extraction energy loss (*Adv. Energy Mater.* 2018,8, 1701159). Additionally, the slow drying process of the carbon paste electrode might induce better contact with the perovskite layer, thus decreasing the series resistance and increasing the J_{sc} . We hope that our responses have addressed your comments. Conducting a more in-depth investigation on the FAPbBr_3 /carbon interface to elucidate the mechanism of the carrier transfer process across the perovskite/carbon interface is beyond the scope of this article.

Changes in the J_{sc} and V_{oc} of perovskite solar cells undeniably impact the performance of photoanodes. However, given that the stability test for the photoanode device falls within a 100-hour range, while the performance evolution of the solar cell device extends to hundreds of days, it's challenging to determine the substantial impact of these device alterations on stability. Nevertheless, according to our stability test results, after 100 hours, the principal cause of photoanode degradation is the water penetration resulting from the failure of the epoxy seal. As such, we are convinced that enhancing encapsulation technology is a practical approach to further improve photoanode stability.

Comments 3: From the same perspective as the previous question, what is the stability measurement method under one-sun condition (MPPT or open circuit condition, etc.)? Also,

during the initial stage, the efficiency tends to increase. What could be the cause of this, and did these characteristics also affect the stability of the photoanode?

Response: We thank the reviewer for pointing out the imprecision in the stability tests of solar cells. The stability measurement tests under one-sun conditions were conducted with open circuit conditions.

The observed performance enhancement during the initial phase of stability testing is largely attributable to the light-soaking effect. This effect manifests in perovskite solar cells and significantly enhances the performance of these cells through sustained light exposure over a given period. Similar observations have been made in silicon solar cells, copper indium gallium selenide (CIGS) solar cells, CdTe solar cells, dye-sensitized solar cells (DSSCs), and polymer solar cells. The light soaking effect in perovskite solar cells primarily originates from the following phenomena (*ACS Appl. Energy Mater.* Article ASAP DOI: 10.1021/acsaem.2c04120):

1. light-induced ion migration,
2. light-induced trap filling or trap deactivation,
3. light-induced lattice expansion, and
4. light-induced fluctuation in charge carrier accumulation.

Overall, the light-soaking effect in solar cell devices is primarily related to changes in mobile charge carriers that exist in bulk or interfaces of functional layers, which include ions and electrons. The precise mechanism of the light soaking effect can vary case by case, and a thorough exploration of this topic extends beyond the scope of the current study.

Concerning the stability of the photoanode, we observed a similar phenomenon during its stability testing (**Figs. 4a** and **4b**). Specifically, the catalytic current tends to incrementally increase in the initial phase, a feature that can be attributed to the light-soaking effect present at the perovskite layer/interfaces. Moreover, we noted that a similar phenomenon was also observed in the CsPbBr₃ photoanode (**Fig. 2 for Reviewer #1**). This suggests that the light-soaking effect might be a common feature in such perovskite photoanode systems.

Fig. 2 for Reviewer #1 Long-term stability of $\text{TiO}_2|\text{CsPbBr}_3|\text{m-c|GS70}$ in 0.1 M KNO_3 electrolyte solution pH 7. (*Nat. Commun.* 10, 2097 (2019))

Comments 4: Graphite sheets are not typically used as a standalone material for preventing for penetration. In Figure 4, the form of degradation appears to be due to water penetration. Even if electrolyte is placed on top of a graphite sheet without operating photoanode, wouldn't it prevent water penetration?

Response: We thank the reviewer for the comment. We maintain that the degradation process depicted in **Fig. 4a** is not attributable to water penetration. To substantiate this, we controlled the exposure area in **Fig. S28a** (in revised supporting information) and observed that degradation only transpires in the area exposed to light. If this process were instigated by water permeation from the catalyst side, the modification in the exposed area should not influence the extent of degradation, i.e., then the entire perovskite layer will be degraded.

Fig. S28 (a) Digital image of selectively exposed photoanode after over 50-hours' stability test.

As demonstrated in **Fig. 4b**, after over 100 hours of testing, the device indeed appears to have degraded due to water penetration. This is evident from the glass side image, which shows noticeable transformations in the perovskite layer. Nevertheless, the degradation process

initiates from the periphery of the perovskite layer. This is attributable to the NiFe side resin glue's deterioration under prolonged exposure to high temperatures and strong alkali conditions (**Fig. S33** of the revised supporting information). The graphite layer on the NiFe catalyst side continues to exhibit excellent water resistance, as indicated by the central part of the perovskite layer remaining proper.

Fig. S33 (a) Schematic illustration of device degradation due to seal failure. (b) Digital images of the photoanode after operation time of over 125 hours. The edge of the device clearly demonstrates the degradation of the perovskite layer due to water penetration.

To assess the waterproof capabilities of graphite sheets, we examined the effect when applied as a layer on a perovskite layer and exposed to water. As shown in **Fig. S16** of the revised supporting information, notable degradation was observed after just 5 minutes of water immersion for the epoxy resin-encapsulated device with solely the mesoporous carbon layer. In contrast, the device protected by the graphite sheets layer exhibited negligible changes even after 100 hours of water immersion. This suggests that a 160 μm layer of stacked graphite sheets is highly effective in impeding water penetration. In the meantime, the graphite sheets serve as an excellent thermal and electrical conductor. Additionally, the self-adhesive nature and commercial availability of the graphite sheets simplify the device fabrication process, rendering them an ideal water-separation component for the assembly of perovskite photoanodes.

Fig. S16 Waterproof performance test of carbon-protected and GS-protected devices. The devices were immersed in pure water for 100 hours.

Comments 5: In line 180~182 and Figure 2e, the ABPE value isn't best of those reported for photoanodes, it should be revised.

Response: The related expression has been revised to a humbler one "To the best of our knowledge, the value is **one of** the best of those reported for photoanodes, including metal oxide, perovskite, polymer bulk heterojunction (BHJ), and silicon-based photoanodes."

Comments 6: It was known that the adhesion between SnO₂ layer and FAPbBr₃ was strong. And authors fabricate FAPbBr₃ photoanode with high adhesive epoxy resin. How the tested sample could be easily separated into two parts?

Response: We apologize for not detailing the stripping process comprehensively in the original manuscript. We have now supplemented this information in the supporting information (**Fig. S29**) for clarity.

Indeed, in the pristine device, the adhesion between the SnO₂ layer and FAPbBr₃ layer is robust, making it challenging to mechanically separate the FAPbBr₃/carbon/graphite sheet layers from

the ITO/SnO₂ surface. However, as illustrated in our article, the SnO₂/perovskite interface degrades after enduring prolonged UV light exposure. Upon applying an external force to remove the surrounding resin glue, we noticed a peeling phenomenon occurring between the glass/SnO₂ layers and FAPbBr₃/carbon/graphite sheet layers. This facilitated the easy separation of the graphite sheet from the glass substrate.

Fig. S29 (a) Schematic diagram of disassembling the tested photoanode. (b) Digital images of disassembled photoanode. When an external force is applied to remove the surrounding resin glue (step 2), a distinctive peeling effect is observed at the interface between the glass/SnO₂ layers and the FAPbBr₃/carbon/graphite sheet layers, which facilitates the easy separation of the graphite sheet from the glass substrate (step 3). This phenomenon is specifically attributed to the degradation occurring at the SnO₂ interface.

Comments 7: In Figure 6 and the corresponding manuscript, authors defined the FAPbBr₃ photoanode as a PVPEC configuration and explained its advantages. However, the PVPEC

system refers to a combination of a photovoltaic (PV) cell and a photoelectrochemical cell (PEC) [1,2], and the FAPbBr₃ photoanode does not consist of a combination of two cells that absorb light; it only has a single light-absorbing layer(perovskite). Therefore, it appears that a different expression should be used to describe it.

[1] M. Bake et al. Metal Halide Perovskites for Solar-to-Chemical Fuel Conversion. Adv. Energy Mater. 10, 1902433 (2020).

[2] Park et al. Water Splitting Progress in Tandem Devices: Moving Photolysis beyond Electrolysis. Adv. Energy Mater. 6, 1600602 (2016).

Response: We thank the reviewer for highlighting the unclear naming, and we have revised the relevant wording accordingly. The rationale behind using PVPEC was to underscore that our PEC device is based on photovoltaic materials, thereby distinguishing it from traditional oxide/nitride PEC devices. To clarify, we've now changed this to 'photovoltaic materials-based PEC (**PVM-PEC**)'.

Reviewer #2 (Remarks to the Author):

The authors have reported a metal-halide perovskite-based photoanode for PEC water oxidations and investigated its physical mechanism using experimental and DFT methods. They proposed the potential of lead halide perovskite-based PEC as a promising approach to achieving efficient, cost-effective, and scalable PEC solar fuel production. Using the DFT calculations, the author suggested that the vacancy structure of Br leads to an n-type property of the FAPbBr₃ film. However, the manuscript needs further revisions to clarify the following points:

Response: Firstly, we sincerely thank the referee for their time, effort, and insights into our manuscript. In particular, the constructive comments and suggestions below were very helpful for us to revise and improve our manuscript. We have studied referee's comments carefully and performed some additional calculations accordingly to address their concerns.

(1) Pb_{Br} can shift the Fermi level penetrating into CBM, why authors suggested that only the vacancy structure of Br leads to an n-type property of the FAPbBr₃ film?

Response: We greatly appreciate the valuable comments provided. While Pb_{Br} can also induce a shift in the Fermi level by penetrating into the CBM, the stability of the Pb antisite Br structure is compromised. In order to elucidate this point, we have conducted calculations to determine the defect formation energies (DFEs) for all these defective types in this study. Further details regarding the calculation procedure can be found in the revised supporting information (**Supplementary Note 3**).

Table S7 exhibits the calculated DFEs for Pb_{Br}, V_{Br}, Br_{Pb}, and V_{Pb}. Given that the experimental conditions in this study correspond to a Pb-rich environment (since the perovskite film was fabricated by a two-step method), the DFE for PbBr is measured to be 0.78 eV, which is higher than the DFE of -0.51 eV for the FAPbBr₃ system with a Br vacancy. This indicates that Pb_{Br} is less stable compared to V_{Br}. Consequently, we propose that V_{Br} is the most plausible structure, as suggested by this research, rather than Pb_{Br}.

Table S7 Formation energies of neutral defects considered in this work. The energies are in eV.

Defect type	ΔH_f (eV)
Pb _{Br}	0.78
V _{Br}	-0.51
Br _{Pb}	3.49
V _{Pb}	0.86

(2) Does the author calculate the defect formation energy to determine the type of defect that can be formed in the metal-halide perovskite? Whether the Br vacancy has the lowest defect

formation energy?

Response: We express our gratitude once again for bringing this to our attention. As evident from **Table S7**, the DFE of the Br vacancy, with a value of -0.51 eV, stands as the lowest among all the defective structures considered.

(3)The author performed spin-polarized DFT calculations. But the density of states (DOS) are not spin polarization. Please provide the spin-polarized DOS.

Response: We thank the reviewer for the revision of this error. In order to obtain a more precise characterization of the electronic and energetic properties, our DOS data were analyzed with the spin-orbit coupling (SOC), due to the heavy atom of Pb in the perovskite. Thus, our calculations were conducted within a non-collinear framework, wherein the magnetic moment can freely rotate in all orientations. As a result of the revision, the term "spin-polarized" has been removed from the "Computational details" section in the revised supporting information.

(4)Fig. S21 does not label the Fermi level.

Response: We thank the reviewer for identifying an omission in the figure. We have labeled the Fermi level in **Fig. S22** (Fig. S21 in the original supporting information) in the revised Supporting Information.

Fig. S22 Calculated partial density of states (PDOS) for X-site vacancy (VBr).

Reviewer #3 (Remarks to the Author):

This article presents FAPbBr₃ halide perovskite photoanodes for PEC water oxidation. Halide perovskites are known to be excellent semiconductors for solar cells, but because they are unstable in water their utilisation in other fields such as aqueous photoelectrochemistry is limited. The authors use a previous method published in Nature Comm 2019 by Poli et al. to protect the solar cell and be able to use it as a photoanode. There are some strengths in the article such as a low onset potential and investigation of expected photothermal effects, as well as systematic characterisation and explanations that make the article enjoyable to read and useful. There are also weaknesses, that I list below as follows:

Response: Firstly, we extend our deep gratitude to the reviewer for their time, commitment, and expertise that have greatly contributed to the improvement of our manuscript. In particular, the constructive comments and suggestions below were very helpful for us to further improve the quality of our manuscript. We have studied referee's comments carefully and modified our manuscript accordingly to address their concerns.

- The word limit of 5000 words is exceeded in 1000 words

Response: We thank the reviewer for the comment. We have moved the details about **Fig. 6** to **Supplementary Note 6** in the supporting information to manage the word count. Now, our manuscript is about 5200 words, and we hope it meets the journal's requirements after the final edits.

- This article follows Poli's method presented in Nature Comm 2019, 2097, where mesoporous carbon layer, adhesive graphite sheet and electrocatalyst were used to protect and use a bromide perovskite for PEC water oxidation. However, the introduction or the Methods omits this relevant citation. Poli's article is cited, but only at the end in a comparison of performances. Poli's article should have been cited early on, otherwise it may be seen as appropriating other people's work

Response: We regret the initial omission of Poli's article in our manuscript, and sincerely apologize for this oversight. We now underscore the significance of this research in both the Introduction and Methods sections of our manuscript. We fully agree that the mesoporous carbon/graphite sheet structure explored in Poli's article represents one of the most effective strategies to protect the absorption layer while preserving high conductivity. Additionally, as emphasized in our manuscript, this approach effectively utilizes the photothermal effect to boost the overall efficiency of the device. We are of the view that this straightforward and efficient strategy deserves broader adoption in the PEC field. Once again, we extend our apologies for the initial oversight.

- The ABPE% measurement appears not correct. ABPE values should be measured in 2-electrode configuration. Measurements in 3-electrode configuration are not ABPE. Please see for example: Nat Energy 2, 16191 (2017) <https://doi.org/10.1038/nenergy.2016.191> or textbooks of PEC

Response: ABPE is generally employed to characterize the photo-response efficiency of a photoelectrode under a specified applied voltage, as can be derived from the equation: $(1.23 - V_{\text{bias}}) \times (j_{\text{light}} - j_{\text{dark}}) / P_{\text{light}}$. As mentioned by the reviewer, in a two-electrode system, V_{bias} signifies the applied potential difference between the cathode (or counter electrode) and the photoanode. In this system, the J-V response of the photoanode is influenced not only by the photoanode's intrinsic quality, but also by the half-reaction performance occurring at the counter electrode (for instance, HER, affected by the activity and loading capacity of the Pt catalyst). Consequently, due to variability in the preparation method and performance of the cathode, we maintain that the J-V curves measured in the two-electrode system, as well as the corresponding ABPE values, are challenging to compare directly with other works.

This limitation is not observed in the three-electrode system, where V_{bias} denotes the potential difference between the photoelectrode and the reference electrode, typically RHE. Doing so would result in an interface measurement and not a device measurement since a bias versus a reference electrode excludes the half-reaction occurring at the counter electrode (textbooks, Photoelectrochemical water splitting : standards, experimental methods, and protocols, New York : Springer, 2013, chapter 2.3.1). In this setup, the J-V response of the photoanode is independent and remains unaffected by the HER activity or loading of the platinum counter electrode. Hence, the calculation of the ABPE value in the three-electrode system has become the prevalent method (**Fig. 1 for Reviewer #3**). For further details, we refer you to the following references. Their ABPE data plots employ V_{RHE} as the x-axis, which implies that the ABPE data is derived under the three-electrode system.

ABPE calculated in a three-electrode system.

Nat Catal 3, 932–940 (2020).

Nat Commun 14, 179 (2023).

Nat Commun 12, 6969 (2021).

Nat Commun 11, 4610 (2020)

Nat Commun 13, 729 (2022).

Nat Commun 10, 3687 (2019).

Nat Commun 7, 13380 (2016).

ABPE calculated in a two-electrode system.

Nat Commun 6, 8769 (2015).

Nat Energy 2, 16191 (2017).

Three-electrode system

Two-electrode system

Fig. 1 for Reviewer #3. ABPEs in previously reported literatures.

- The low onset potential is great but is practically attributed to the quality of the semiconductor FAPbBr₃, which is known. Such quality is necessary but not sufficient. Unlike solar cells, PEC devices performance strongly depend on the catalytic performance. If there is no catalysis, there is no photocurrent. The good translation from solar cell photocurrents to PEC photocurrents rely on the quality of the method to protect and extract the charges to the electrolyte, and this is achieved because Poli's method of using mesoporous carbon and graphite sheets functionalised with electrocatalyst is used. This is not properly conveyed, emphasized, and studied in the article to explain the good onset potential

Response: We extend our apologies once more for not sufficiently emphasizing Poli's work initially. We have now revisited this pivotal study in both the Introduction and the main text of the manuscript, underscoring the importance of Poli's methodology.

To further highlight the waterproof capabilities of graphite sheets, we examined the effect when applied as a layer on a perovskite layer and exposed to water. As shown in **Fig. S16** of the revised supporting information, notable degradation was observed after just 5 minutes of water immersion for the epoxy resin-encapsulated device with solely the mesoporous carbon layer. In contrast, the device protected by the graphite sheets layer exhibited negligible changes even after 100 hours of water immersion. This suggests that a 160 μm layer of stacked graphite sheets is highly effective in impeding water penetration. In the meantime, the graphite sheets serve as an excellent thermal and electrical conductor. Additionally, the self-adhesive nature and commercial availability of the graphite sheets simplify the device fabrication process, rendering them an ideal water-separation component for the assembly of perovskite photoanodes.

Fig. S16 Waterproof performance test of carbon-protected and GS-protected devices. The devices were immersed in pure water for 100 hours.

- The use of bromide perovskite, mesoporous carbon and graphite sheet with a NiFe-based electrocatalyst and >100 h stability has already been presented in conferences and literature (DOI: 10.26434/chemrxiv-2023-cl4m5) and should be cited

Response: We are grateful to the reviewers for highlighting the missing citations in our original manuscript. The papers specified have now been appropriately cited and compared in our revised submission (**Fig. 2e, 2f** and **Table 1**).

- The introduction generates expectations on bias-free PEC devices but these are later not presented in the article. The measurements are always in 3-electrode configuration

Response: We thank the reviewer for pointing out the imprecision in the Introduction, and the related narrative in page 3 has been revised for better clarity and accuracy.

- The introduction also generated expectations about the use of the solar broad spectrum, but later the UV part of the solar spectrum can't be used because it degrades the organic part of the perovskite.

Response: We thank the reviewer for pointing out the imprecision in the Introduction, and the related narrative in page 3 has been modified for better clarity and accuracy.

- Page 4: sentence "Additionally, its inorganic counterparts require high-temperature annealing over 300 °C for stabilization, which leads to poor crystallinity in the films.¹²" This needs rewording. It sounds like if crystallinity decreases with temperature, which is the other way around.

Response: We appreciate the reviewer for pointing out the above errors, the description has been corrected.

- Figure 1d shows that the VB edge of FAPbBr₃ is not suitable for water oxidation, contrary to what is shown later in Figure 3. The difference may be in the different pH values, but this may confuse readers if not properly explained

Response: We thank the reviewer for highlighting the deficiencies in our explanation of the band structure, and we now change the water splitting potentials in **Fig. 1c** and **1d** to values at alkaline pH (13.6).

Fig. 1 (a) Schematic illustration of FAPbBr₃ photoanode for oxygen evolution. (b) Comparison of the LSV curves between the FAPbBr₃ photoanode and the best-reported photoanodes made from a single junction absorber layer, including TiO₂, WO₃, Fe₂O₃, BiVO₄, and Ta₃N₅, with catalysts layer. The LSV data was extracted from the corresponding literature. Band positions of (c) typical PEC and (d) lead halide perovskite semiconductors in the pH 13.6 aqueous electrolyte compared with the energy potential for water splitting reaction. red = valence band edge; blue = conduction band edge. Tabulated values are retrieved from reference.

- Sentences "In our study, the theoretical oxidation potential of Br⁻ is 1.087 V vs. NHE (pH = 0), which is equivalent to -5.59 eV vs. vacuum level.³⁰ Under visible light irradiation, the potential is almost the same as the VBM of perovskite. Thus, self-oxidation is less likely to occur on FAPbBr₃ because the reaction requires a certain additional driving force." There are different things here that are not clear. An oxidation happening at 1.087 V vs. NHE (pH = 0) is said not likely, but water oxidation at 1.23 V vs NHE pH 0 (Fig 1) is reported. Moreover, it is not clear what self-oxidation they are referring to.

Response: We appreciate the reviewers for highlighting ambiguities in our discussion, and we have rephrased the descriptions to more clearly articulate our points.

All the electrochemical tests were conducted under strong alkaline conditions, implying that the water oxidation potential at pH 0 is not applicable in this instance. Since protons are not implicated in the oxidation of Br⁻, the oxidation potential of Br⁻ (1.087 V vs. NHE) does not shift with pH. Therefore, the correct formulation is that the theoretical water oxidation potential at, for instance, pH 13.6, is $1.23 - 0.059 \times 13.6 = 0.428$ V vs. NHE, while the theoretical Br⁻ oxidation potential remains at 1.087 V vs. NHE. This observation aligns with our hypothesis that Br⁻ in the perovskite film does not spontaneously oxidize. However, when the perovskite film comes into contact with layers of n-type wide bandgap TiO₂ or SnO₂, highly oxidative holes are generated on the n-type semiconductor's surface. These possess sufficient potential to oxidize the Br⁻, to bromine or free radicals, in the perovskite layer in contact with it, resulting in the degradation of the perovskite layer.[1]

[1] The Journal of Physical Chemistry C 2014 118 (30), 16995-17000.

- Table 1, CsPbBr₃ PEC devices are claimed to last 17h but they lasted 35h in ref. 27

Response: We thank the reviewer for pointing out our mistake. The stability data have been corrected.

- The use of lower and higher CB and VB edge is confusing. Shallower and deeper would be less confusing.

Response: We thank the reviewer for the suggestion, and the relevant descriptions in the main text and supplementary notes have been modified accordingly.

REVIEWERS' COMMENTS

Reviewer #1 (Remarks to the Author):

The authors revised this manuscript well. But, I am strongly against ">100 hr stability" in the title.

There are two problems.

- 1) ~50% of initial photocurrent after 120hr cannot mean "stable"
- 2) The authors used "UV filters". It is not conventional condition for PEC experiment even some researchers reported via this way.

So, I can only accept this manuscript with change of the title. I recommend

- 1) remove 100 hr value from title. Just "enhanced stability" is enough for the data.

This is due to that the title misleads other researchers.

Reviewer #2 (Remarks to the Author):

Authors have addressed all my concerns, I recommend the manuscript for publication.

Reviewer #3 (Remarks to the Author):

I thank the authors for addressing most comments.

I agree that ABPE values in three-electrode systems are different (and higher) than in two electrode systems. I also agree that we can find both approaches in the literature, although the first and original ABPE measurements were done in two-electrode systems. To avoid disagreements in the comparisons of ABPE with literature, the authors should explicitly cite that ABPE measurements are done in a 3-electrode system everywhere these values are discussed to avoid any confusion. This would be a constructive solution to this situation where ABPE values in literature are measured in two different ways.

Page 10-11 says "the photovoltage of photoanodes is calculated as the difference between the photocurrent onset potential and the Nernstian potential for water oxidation (1.23 V vs. RHE)". However, this ignores the overpotential to drive water oxidation by the electrocatalyst. Figure 5o shows around 250 mV of overpotential. How is this accounted for in the photoanode photovoltage?

Related to previous one, the photoanode onset potential is around -0.1 V vs RHE (Fig. 2b), so I understand that this makes a photovoltage of $1.23+0.25-(-0.1)=1.58$ V . However, Figure S2d shows a Voc of 1.39. How can the photovoltage as a photoanode be larger than the Voc as a solar cell?

The authors claim over 100 hours stability; however, this was reached in two parts with a 12 hours long recovery after 72 hours of operation. Why it degrades ? And what happens during the recovery? Some explanations in the manuscript on these aspects are needed. This needed recovery should be mentioned in the Table when the performance is compared to other photoanodes that were not allowed a recovery...

REVIEWER COMMENTS

Reviewer #1 (Remarks to the Author):

The authors revised this manuscript well. But, I am strongly against ">100 hr stability" in the title. There are two problems.

1) ~50% of initial photocurrent after 120hr cannot mean "stable"

2) The authors used "UV filters". It is not conventional condition for PEC experiment even some researchers reported via this way.

So, I can only accept this manuscript with change of the title. I recommend 1) remove 100 hr value from title. Just "enhanced stability" is enough for the data.

This is due to that the title misleads other researchers.

Response: We are grateful to the reviewer for their insightful suggestions regarding the title of the manuscript. We agree with the reviewer's viewpoints; hence, as suggested, we have revised the title to "Monolithic FAPbBr₃ Photoanode for Photoelectrochemical Water Oxidation with Low Onset-potential and Enhanced Stability".

Reviewer #2 (Remarks to the Author):

Authors have addressed all my concerns, I recommend the manuscript for publication.

Response: We appreciate the reviewer for their encouraging feedback.

Reviewer #3 (Remarks to the Author):

I thank the authors for addressing most comments.

I agree that ABPE values in three-electrode systems are different (and higher) than in two electrode systems. I also agree that we can find both approaches in the literature, although the first and original ABPE measurements were done in two-electrode systems. To avoid disagreements in the comparisons of ABPE with literature, the authors should explicitly cite that ABPE measurements are done in a 3-electrode system everywhere these values are discussed to avoid any confusion. This would be a constructive solution to this situation where ABPE values in literature are measured in two different ways.

Response: We fully agree with the reviewers' comments. For clarity, we have updated the main text and methods parts to specify that the ABPE values were derived using a three-electrode system.

Page 10-11 says “the photovoltage of photoanodes is calculated as the difference between the photocurrent onset potential and the Nernstian potential for water oxidation (1.23 V vs. RHE)”. However, this ignores the overpotential to drive water oxidation by the electrocatalyst. Figure 5o shows around 250 mV of overpotential. How is this accounted for in the photoanode photovoltage?

Response: We thank the reviewer for these additional comments. For the sake of consistency when comparing with other PEC systems, we refer to the literature [1, 2] to define photovoltage as the difference between the photocurrent onset potential and the Nernstian potential for water oxidation. Notably, this approach neglects catalytic overpotentials, resulting in a potential underestimation of the photovoltage for PEC devices—particularly for photoanodes given the substantial overpotentials for water oxidation.

For many PEC systems, the direct determination of the water oxidation overpotential for the electrocatalyst is unfeasible owing to constraints in their fabrication methods. Take, for instance, the in-situ growth of metal oxide catalysts on mesoporous bismuth vanadate surfaces. Given that the specific loading surface area and capacity remain indeterminate, independently determining the overpotential requirements of the electrocatalyst layer(s) becomes challenging. Concurrently, not all published PEC systems provide data on the catalytic performance of the electrocatalyst layer(s), rendering the exact photovoltage calculation unattainable on our end. In this context, we've opted for this practical approach. The photovoltage here can be viewed as an alternative interpretation of the onset potential. Its core purpose is to assess the onset potential's magnitude, positioning 1.23 V as a standardized reference across diverse PEC systems.

Prompted by the reviewers' comments, we've underscored the limitations of this approach within the methods section. We hope that our responses have addressed your concerns. Nonetheless, it's evident that our PEC devices show notably enhanced photovoltage, especially compared to devices with wide-band gap absorbers.

[1] Current Opinion in Electrochemistry, 2017, 2(1), 104-110.

[2] Energy Environ. Sci., 2015, 8, 2886-2901

Related to previous one, the photoanode onset potential is around -0.1 V vs RHE (Fig. 2b), so I understand that this makes a photovoltage of $1.23+0.25-(-0.1)=1.58$ V . However, Figure S2d shows a V_{oc} of 1.39. How can the photovoltage as a photoanode be larger than the V_{oc} as a solar cell?

Response: We thank the reviewer for these additional comments. At an overpotential of 0.25 V, the catalysis current exceeds 20 mA cm^{-2} , as presented in **Supplementary Fig. 9b**. Therefore, this value should not be utilized to determine the photovoltage. As demonstrated in **Fig. 1a for Reviewer #3**, the actual photovoltage is determined by comparing the difference between the J - V curves of the photoanode and the electrocatalysts at an identical scan rate. The photovoltage

is conclusively established at 1.364 V for a current density of 1.0 mA cm⁻² and 1.335 V for a current density of 5.0 mA cm⁻². It is imperative to highlight that this calculation omits considerations like the reduction in the theoretical potential of water oxidation and the enhancement of catalytic performance due to the photothermal effect of the photoanode (an additional overpotential reduction of at least 20 mV can be achieved). Hence, the FAPbBr₃ device, with an open circuit voltage of 1.4 V, can fully deliver an apparent anodic current before reaching 0 V vs. RHE. It should be acknowledged that the current at the foot of the wave area might be due to Ni^{2+/3+} oxidation rather than water oxidation. This also explains the sharp current peaks observed under the chopping light conditions at this potential range (**Fig. 2c**). According to the Butler-Volmer equation, the catalytic current exhibits exponential growth. Thus, we deduced the onset potential of the catalytic current for water oxidation using a logarithmic scale. As shown in **Fig. 1b for Reviewer #3**, the NiFe catalysts can theoretically achieve a catalytic current of 1.0 mA cm⁻² at 1.38 V vs. RHE, which implies that ‘real’ water oxidation onset potential on photoanode may still be around 0V, a value superior to many PEC systems.

Fig. 1 for Reviewer #3. (a) The comparison for *J-V* curves of the photoanode and the electrocatalysts. (b) *J-V* curve of the electrocatalysts that is plotted in logarithmic coordinates, the ideal water oxidation onset (reach a current density of 1.0 mA cm⁻²) is determined to be 1.38 V.

The authors claim over 100 hours stability; however, this was reached in two parts with a 12 hours long recovery after 72 hours of operation. Why it degrades ? And what happens during the recovery? Some explanations in the manuscript on these aspects are needed. This needed recovery should be mentioned in the Table when the performance is compared to other photoanodes that were not allowed a recovery...

Response: We thank the reviewer for the comment. The reversible losses, characterized by efficiency gains obtained by re-measuring after dark recovery, are commonly observed in n-i-p

structured PSCs (**Fig. 2 for Reviewer #3**).[1-5] Meanwhile, as shown in **Fig. 2b for Reviewer #3**, this dark recovery strategy can effectively improve the stability of perovskite devices, so we suggest that this strategy can be applied in evaluating perovskite PEC devices.

The reversible losses and dark recovery phenomenon are often attributed to ion and defect migration within the perovskite layer and at the interfaces between perovskite and selective contacts. While the exact mechanism may differ in specific cases, a comprehensive investigation of this subject falls outside the scope of this study.

Fig. 2 for Reviewer #3. (a) Aging perovskite devices show significant efficiency recovery after resting in the dark. (b) The effect of light cycling on aging.

To preclude potential misunderstandings, we have substituted the term "100h" with "Enhanced" in the title and incorporated relevant annotations in **Table 1**. In accordance with the reviewer's recommendation, we have provided a concise explanation of this phenomenon in the relevant section of the main text.

- [1] Joule, 2018, 2(6), 1019-1024.
- [2] Science, 2017, 355(6326): 722-726.
- [3] Science, 2016, 354(6309): 206-209.
- [4] Nat Energy, 2018, 3(1), 61-67.
- [5] Nat Energy, 2020,5(1), 35-49.